# Structural basis for the hijacking of endosomal sorting nexin proteins by *Chlamydia trachomatis*

**Blessy Paul[1], Hyun Sung Kim[1], Markus C Kerr[1], Wilhelmina M Huston[2], Rohan D Teasdale[1]\*, Brett M Collins[1]\***

[1]Institute for Molecular Bioscience, The University of Queensland, St. Lucia, Australia; [2]School of Life Sciences, University of Technology Sydney, Sydney, Australia

**Abstract** During infection chlamydial pathogens form an intracellular membrane-bound replicative niche termed the inclusion, which is enriched with bacterial transmembrane proteins called Incs. Incs bind and manipulate host cell proteins to promote inclusion expansion and provide camouflage against innate immune responses. Sorting nexin (SNX) proteins that normally function in endosomal membrane trafficking are a major class of inclusion-associated host proteins, and are recruited by IncE/CT116. Crystal structures of the SNX5 phox-homology (PX) domain in complex with IncE define the precise molecular basis for these interactions. The binding site is unique to SNX5 and related family members SNX6 and SNX32. Intriguingly the site is also conserved in SNX5 homologues throughout evolution, suggesting that IncE captures SNX5-related proteins by mimicking a native host protein interaction. These findings thus provide the first mechanistic insights both into how chlamydial Incs hijack host proteins, and how SNX5-related PX domains function as scaffolds in protein complex assembly.

**\*For correspondence:** r.teasdale@ imb.uq.edu.au (RDT); b.collins@ imb.uq.edu.au (BMC)

**Competing interests:** The authors declare that no competing interests exist.

## Introduction

To counter host defence mechanisms intracellular bacterial pathogens have evolved numerous strategies to evade immune detection, replicate and cause infection. Many pathogens manipulate endocytic pathways to gain entry into host cells and generate a membrane-enclosed replicative niche. This frequently involves hijacking or inhibiting the host cell trafficking machinery, first to generate the pathogen containing vacuole (PCV) and subsequently to prevent fusion with lysosomal degradative compartments. Concomitantly the pathogen often endeavors to decorate the PCV with host proteins and lipids that mimic other host cell organelles in order to circumvent innate immune detection, expand the replicative niche and acquire nutrients to support intracellular replication (*Di Russo Case and Samuel, 2016*; *Personnic et al., 2016*). This process is often orchestrated through the action of molecular syringe-like secretion systems that deliver bacterial effector proteins directly into the host cell cytoplasm.

*Chlamydia trachomatis* is arguably one of the most successful human bacterial pathogens by virtue of its capacity to hijack host cell intracellular trafficking and lipid transport pathways to promote infection (*Bastidas et al., 2013*; *Derré, 2015*; *Elwell et al., 2016*; *Moore and Ouellette, 2014*). *C. trachomatis* causes nearly 100 million sexually transmitted infections annually worldwide, and if left unchecked leads to various human diseases including infection-induced blindness, pelvic inflammatory disease, infertility and ectopic pregnancy (*Aral et al., 2006*; *Newman et al., 2015*). Even though chlamydial infections can generally be treated with antibiotics, persistent infections remain a challenge (*Kohlhoff and Hammerschlag, 2015*; *Mpiga and Ravaoarinoro, 2006*).

**eLife digest** The bacterium *Chlamydia trachomatis*, commonly known as chlamydia, is a frequent cause of sexually transmitted infections, and a leading cause of blindness due to infection. The bacteria must directly enter the cells of its human host to grow and multiply. Inside a human cell, the bacteria form and then develop within specialized compartments called inclusions that are surrounded by membrane. The outside of the inclusion membrane becomes coated with dozens of unique bacterial proteins. The major role of these bacterial proteins is to hijack other proteins in the human cell to generate and maintain the membrane of the inclusion compartments.

One bacterial protein in particular, called IncE, is able to bind to specific host proteins called sorting nexins. These host proteins normally control the formation of tube-like membrane structures, which transport fatty molecules and proteins throughout the cell. The IncE protein is thought to recruit sorting nexins to help shape the inclusion membrane and perhaps control which types of proteins and fatty molecules associate with it. However, until now it was unknown how IncE, or any similar protein for that matter, could specifically hijack a host cell protein.

Now, Paul et al. have revealed the three-dimensional structure of a human sorting nexin protein, called SNX5, bound to a small fragment of the IncE protein from chlamydia. The structure shows that the part of SNX5 that associates with IncE is the part of the protein normally thought to interact with specific fatty molecules rather than proteins. Further experiments showed that SNX5 was still recruited to the inclusion compartment when the amount of these fatty molecules in human cells was reduced. However, this was not the case if SNX5 was prevented from interaction with the IncE protein.

Paul et al. also observed that the site on SNX5 where IncE binds is almost identical in related proteins from many other species, including zebrafish and worms, most of which are not hosts for chlamydia. This lead them to suspect that IncE hijacks the sorting nexin proteins by mimicking an important host protein that is yet to be discovered.

Proteins in the inclusion membrane play many important roles, and so this work on IncE only provides the first glimpse at how these proteins are able to manipulate the machinery of the host cell to their own ends. Further studies will therefore be needed to understand how these proteins exploit their host environment at the molecular level, and might be targeted in new antibacterial approaches. The findings also show how studying bacteria that live within host cells, like chlamydia, can provide insight into how other molecules are normally transported within cells: a process that is fundamental to all living cells.

All *Chlamydiae* share a common dimorphic life cycle, where the bacteria alternates between the infectious but non-dividing elementary body (EB) form, and the non-infectious but replicative reticulate body (RB) form. Following internalization of EBs through a poorly defined endocytic process, the bacteria reside in a membrane-bound vacuole termed the inclusion, where they convert into RBs and replication occurs over 24–72 hr. RBs eventually redifferentiate back to EBs in an asynchronous manner, and are then released to infect neighboring cells (*Di Russo Case and Samuel, 2016*; *Hybiske, 2015*; *Ward, 1983*). The encapsulating inclusion membrane provides the primary interface between the bacteria and the host cell's cytoplasm and organelles. From the initial stages of invasion until eventual bacterial egress the chlamydial inclusion is extensively modified by insertion of numerous Type-III secreted bacterial effector proteins called inclusion membrane proteins or 'Incs'. The Incs modulate host trafficking and signaling pathways to promote bacterial survival at different stages, including cell invasion, inclusion membrane remodeling, avoidance of the host cell innate immune defense system, nutrient acquisition and interactions with other host cell organelles (*Elwell et al., 2016*; *Moore and Ouellette, 2014*; *Rockey et al., 2002*).

*Chlamydiae* secrete more than fifty different Inc proteins. While Incs possess little sequence similarity, they share a common membrane topology with cytoplasmic N- and C-terminal domains, separated by two closely spaced transmembrane regions with a short intra-vacuolar loop (*Dehoux et al., 2011*; *Kostriukova et al., 2008*; *Li et al., 2008*; *Lutter et al., 2012*; *Rockey et al., 2002*) (*Figure 1A*). The cytoplasmic N- and C-terminal sequences of the Inc proteins act to bind and

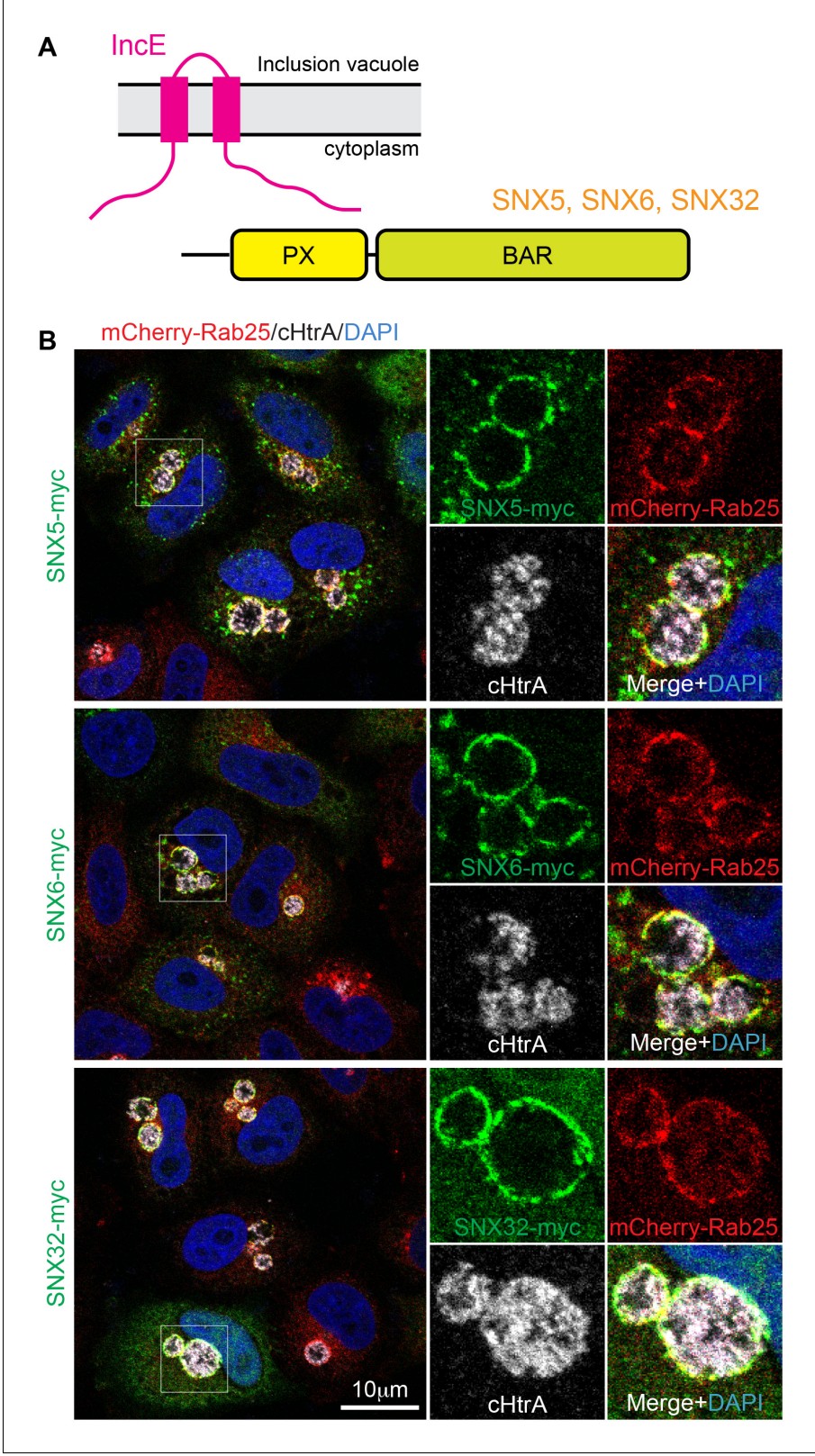

**Figure 1.** SNX5, SNX6 and SNX32 tare recruited to *C. trachomatis* inclusions. (**A**) HeLa cells stably expressing the mCherry-Rab25 inclusion membrane marker (red) were infected with *C. trachomatis* serovar L2 (24 hr) and transfected with myc-tagged SNX expression constructs. The samples were fixed and immunolabeled with anti-

*Figure 1 continued on next page*

*Figure 1 continued*

myc (green) and anti-chlamydial HtrA antibodies (white) and counterstained with DAPI (blue). Similar experiments using GFP-tagged proteins are shown in *Figure 1—figure supplement 1A*.

The following figure supplements are available for figure 1:

**Figure supplement 1.** SNX5, SNX32 and SNX1 are recruited to *C. trachomatis* inclusions and membrane tubules.

**Figure supplement 2.** Recruitment of SNX1, SNX2 and SNX5 to inclusions is not dependent on 3-phosphoinositides.

manipulate host cell proteins. Reported examples include the binding of the small GTPase Rab4A by CT229 (*Rzomp et al., 2006*), Rab11A by Cpn0585 (*Cortes et al., 2007*), SNARE proteins by IncA (*Delevoye et al., 2008*), centrosomal and cytoskeletal proteins by Inc850 and inclusion protein acting on microtubules (IPAM) (*Dumoux et al., 2015*; *Mital et al., 2015*, *2010*), myosin phosphatase by CT228 (*Lutter et al., 2013*), 14-3-3 and Arf family proteins by IncG and InaC (*Kokes et al., 2015*; *Scidmore and Hackstadt, 2001*), and the lipid transfer protein CERT by IncD (*Derré et al., 2011*; *Elwell et al., 2011*). Despite these reports, there are no known structures of Inc family members either alone or in complex with host effectors.

Two recent studies have greatly expanded the repertoire of host cell proteins known to associate with chlamydial inclusions and Inc proteins (*Aeberhard et al., 2015*; *Mirrashidi et al., 2015*). Both reports confirmed that membrane trafficking proteins are major components of the inclusion proteome; and in particular members of the endosomal sorting nexin (SNX) family are highly enriched. Specifically it was shown that the *C. trachomatis* IncE/CT116 protein could recruit SNX proteins containing bin-amphiphysin-Rvs (BAR) domains SNX1, SNX2, SNX5 and SNX6 (*Mirrashidi et al., 2015*). SNX1 and SNX2 are highly homologous and form heterodimeric assemblies with either SNX5 or SNX6 to promote endosomal membrane tubulation and trafficking (*van Weering et al., 2012*). A fifth protein SNX32 is highly similar to SNX5 and SNX6 but is almost exclusively expressed in the brain and has not yet been characterized. SNX recruitment to the inclusion occurs via the C-terminal region of IncE interacting with the phox-homology (PX) domains of SNX5 or SNX6 (*Mirrashidi et al., 2015*) (*Figure 1A*). Interestingly, RNAi-mediated depletion of SNX5/SNX6 does not slow infection but rather increases the production of infectious *C. trachomatis* progeny suggesting that the SNX recruitment is not done to enable bacterial infection. Instead it was proposed that because SNX proteins regulate endocytic and lysosomal degradation, the manipulation by IncE could be an attempt to circumvent SNX-enhanced bacterial destruction (*Aeberhard et al., 2015*; *Mirrashidi et al., 2015*).

Here we use X-ray crystallographic structure determination to define the molecular mechanism of SNX5-IncE interaction, and confirm this mechanism using mutagenesis both in vitro and in cells. When bound to SNX5, IncE adopts an elongated $\beta$-hairpin structure, with key hydrophobic residues docked into a complementary binding groove encompassing a helix-turn-helix structural extension that is unique to SNX5, SNX6, and the brain-specific homologue SNX32. A striking degree of evolutionary conservation in the IncE-binding groove suggests that IncE co-opts the SNX5-related molecules by displacing a host protein (as yet unidentified) that normally binds to this site. Our work thus provides both the first mechanistic insights into how protein hijacking is mediated by inclusion membrane proteins, and also sheds light on the functional role of the SNX5-related PX domains as scaffolds for protein complex assembly.

## Results

### IncE specifically binds and recruits SNX5, SNX6 and SNX32 to *C. trachomatis* inclusions

It was previously shown that the sorting nexins SNX1, SNX2, SNX5 and SNX6 are recruited to the inclusion membrane in *C. trachomatis* infected cells (*Aeberhard et al., 2015*; *Mirrashidi et al., 2015*). We first confirmed this for myc-tagged SNX1, SNX2 and SNX5 in HeLa cells infected with *C.*

*trachomatis* serovar L2 (MOI ~0.5) for 18 hr. All three proteins were recruited to the inclusion membrane as assessed by co-localisation with the inclusion marker mCherry-Rab25 (*Figure 1B*) (*Teo et al., 2016*), as were GFP-tagged SNX1 and SNX5 but not the more distantly homologous GFP-SNX8 (*Figure 1—figure supplement 1A*). We also observed localization of the SNX proteins to extensive inclusion-associated membrane tubules in a subset of infected cells as described previously (*Figure 1—figure supplement 1B*) (*Aeberhard et al., 2015*; *Mirrashidi et al., 2015*). Interestingly, when infected cells are treated with wortmannin, a pan-specific inhibitor of phosphoinositide-3-kinase (PI3K) activity, we see a loss of the SNX proteins from punctate endosomes, but not from the inclusion membrane (*Figure 1—figure supplement 2*; *Video 1*). A similar result is seen for specific inhibition of PtdIns3*P* production by Vps34 using Vps34-IN1 (*Figure 1—figure supplement 2*). This offers two possibilities; that either SNX recruitment to the inclusion occurs via protein-protein interactions, and does not depend on the presence of 3-phosphoinositide lipids that typically recruit SNX proteins to endosomal membranes, or alternatively that PI3Ks are not present at the inclusion and therefore wortmannin treatment has no effect at this particular compartment. Given our structural and mutagenesis studies below we favor the former explanation.

*Mirrashidi et al. (2015)*, demonstrated an in vitro interaction between IncE and the SNX5 and SNX6 PX domains. To confirm their direct association we assessed the binding affinities using isothermal titration calorimetry (ITC) (*Figure 2A*; *Table 1*). Initial experiments with the human SNX5 and SNX6 PX domains showed robust interactions with the IncE C-terminal domain (residues 107–132). The affinities ($K_d$) for SNX5 and SNX6 were essentially indistinguishable (0.9 and 1.1 µM respectively), but we detected no interaction with the PX domain of SNX1 confirming the binding specificity. The PX domains of SNX5 and SNX6 possess a helix-turn-helix structural insert (*Koharudin et al., 2009*), which is not found in any other SNX family members except for SNX32 (*Figure 2B*), a homologue expressed primarily in neurons (BioGPS (*Wu et al., 2009*)). Confirming a common recruitment motif in the SNX5-related proteins, ITC showed a strong interaction between IncE and the SNX32 PX domain similar to SNX5 and SNX6 ($K_d$ = 1.0 µM) (*Figure 2A*; *Table 1*), and SNX32 was robustly recruited to inclusion membranes in infected cells (*Figure 1B*; *Figure 1—figure supplement 1A*). Overall, our data indicates that a common structure within the SNX5, SNX6 and SNX32 PX domains is required for IncE interaction.

Finally we tested a series of IncE truncation mutants for their binding to the SNX5 PX domain (*Figure 3A, B and C*; *Table 2*). Synthetic peptides were used with single amino acids removed sequentially from the N and C-terminus to determine the minimal sequence required for binding. These experiments showed that the shortest region of IncE able to support full binding to SNX5 encompasses residues 110–131 (GPA VQFFKGKNGSADQVILVT), while a shorter fragment containing residues 113–130 (VQFFKGKNG SADQVILV) can bind to SNX5 with a slightly reduced affinity. While variations are observed across the different *C. trachomatis* serovars (*Harris et al., 2012*) the SNX5-binding sequence appears to be preserved in all detected variants (*Figure 3D*). A comparison with other chlamydial species suggests that IncE is not very widely conserved in this Genus, being clearly identifiable only in the closely related mouse pathogen *C. muridarum* and swine pathogen *C. suis* (*Figure 2C*). Residues required for binding to SNX5 are preserved in these IncE homologues, but whether SNX proteins are also recruited

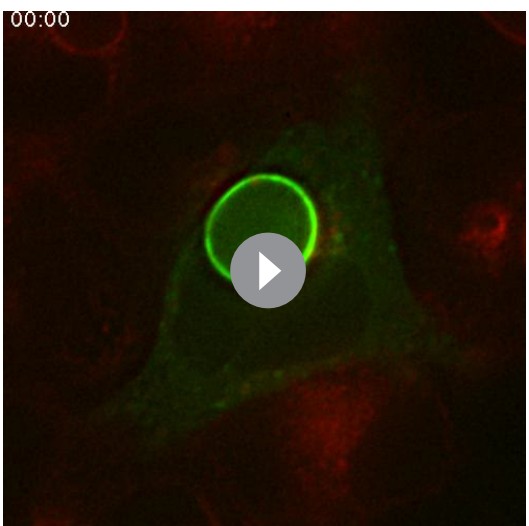

**Video 1.** Movie showing that wortmannin disrupts SNX5 recruitment to endosomes but not the chlamydial inclusion. HeLa cells stably expressing mCherry-Rab25 (red) were transfected transiently with GFP-SNX5 (green) and infected with *Chlamydia trachomatis* L2 for 24 hr. Time-lapse videomicroscopy was performed using an interval of 1 min on an inverted Nikon Ti-E deconvolution microscope with environmental control at 40 x magnification. 10 min into recording 100 nM wortmannin was added.

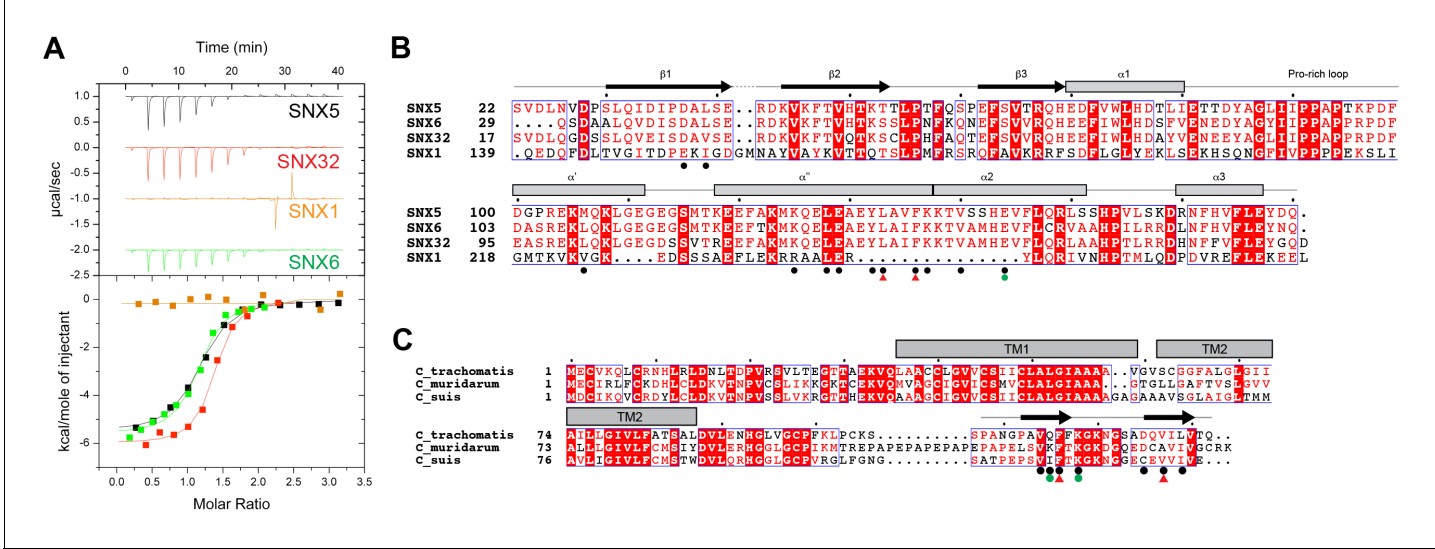

**Figure 2.** IncE from C. trachomatis binds the PX domains of SNX5, SNX6 and SNX32. (A) Binding affinity between IncE peptide (residues 107–132) and SNX PX domains by ITC. Top panels show raw data and lower panels show normalised integrated data. See *Table 1* for the calculated binding parameters. Truncation analyses of the IncE peptide by ITC are shown in *Figure 3*, *Table 2*. (B) Sequence alignment of human SNX1, SNX5, SNX6 and SNX32 PX domains. Conserved residues are indicated in red. Side-chains that directly contact IncE in the crystal structure are indicated with black circles. Mutations that block IncE binding are highlighted with red triangles, and mutations that do not affect binding indicated with green circles. Secondary structure elements derived from the SNX5 crystal structure are indicated above. (C) Sequence alignment of IncE from *C. trachomatis* and putative homologues from *C. muridarum* and *C. suis*. IncE side-chains that directly contact SNX5 in the crystal structure are indicated with black circles. Mutations that block SNX5 binding are highlighted with red triangles, and mutations that do not affect binding indicated with green circles. Predicted transmembrane regions are indicated and C-terminal IncE sequences that form β-strands in complex with SNX5 are shown.

during infection by these other chlamydial species remains to be determined.

## The crystal structure of IncE in complex with the SNX5 PX domain

The canonical PX domain structure is composed of a three-stranded β-sheet (β1, β2 and β3) followed by three close-packed α-helices. The first and second α-helices are connected by an extended proline-rich sequence. Typically PX domains have been found to bind to the endosome-enriched lipid phosphatidylinositol-3-phosphate (PtdIns3P) via a basic pocket formed at the junction between the β3 strand, α1 helix and Pro-rich loop. In contrast SNX5, SNX6 and SNX32 possess major alterations in the PtdIns3P-binding pocket that preclude canonical lipid head-group docking (see below). In addition they possess a unique extended helix-turn-helix insert between the Pro-rich loop and α2 helix of unknown function (*Figure 2B*) (*Koharudin et al., 2009*).

**Table 1.** Thermodynamic parameters of IncE binding to SNX PX domains*.

| Sample cell | Titrant | $K_d$ (μM) | $\triangle H$ (kcal/mol) | $T\triangle S$ (kcal/mol) | $\triangle G$ (kcal/mol) | N |
|---|---|---|---|---|---|---|
| SNX5 PX | IncE peptide[†] | 0.95 ± 0.07 | −6.9 ± 0.3 | −1.9 ± 0.05 | −8.2 ± 0.01 | 1.01 ± 0.01 |
| SNX6 PX | IncE peptide | 1.13 ± 0.08 | −5.0 ± 0.9 | −3.0 ± 1 | −8.0 ± 0.07 | 1.01 ± 0.08 |
| SNX32 PX | IncE peptide | 1.15 ± 0.07 | −6.9 ± 0.4 | −1.3 ± 0.8 | −8.2 ± 0.4 | 1.06 ± 0.005 |
| SNX1 PX | IncE peptide | No binding | | | | |

*Values are the mean from three experiments ±SEM.
b.[†]IncE synthetic peptide sequence PANGPAVQFFKGKNGSADQVILVTQ.

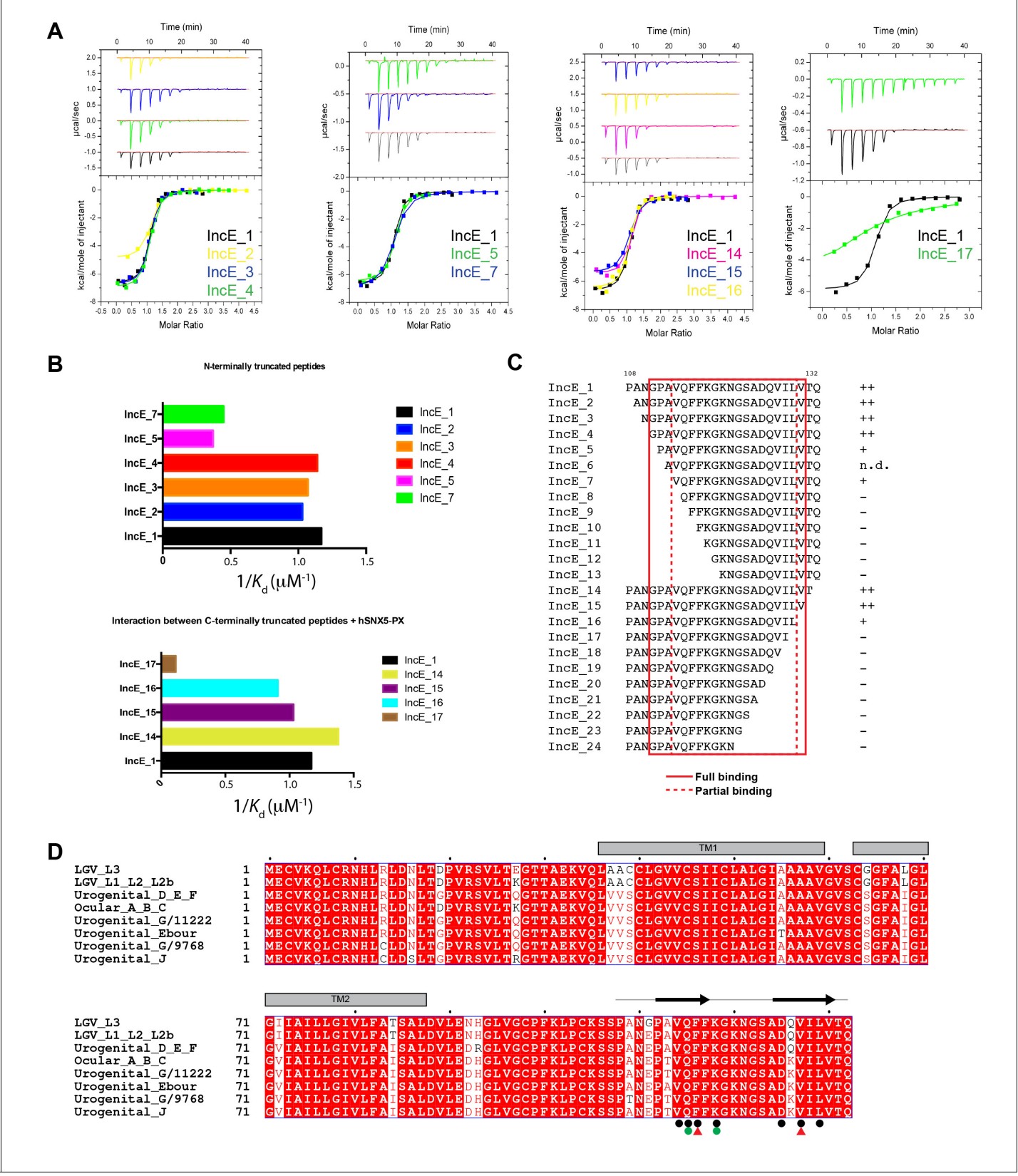

**Figure 3.** IncE residues 110–131 are sufficient for full recognition of the SNX5 PX domain. (**A**) Representative ITC experiments for truncated IncE peptides. These experiments were conducted using a single batch of SNX5 PX domain to minimize batch-to-batch protein variation. (**B**) Plots of the
*Figure 3 continued on next page*

*Figure 3 continued*

affinity constants for selected peptides to highlight the progressive loss of binding with N and C-terminal truncations. (**C**) Sequences of the truncated IncE peptides are given, with a qualitative indication of binding strength relative to the IncE_1 peptide containing residues 107–132. Full binding is indicated by '++' reduced binding by '+' and lack of binding by '−'. All sequence information and their $K_d$ values are given in **Table 2.** When compared to the reference ITC experiment the binding affinity of peptides was unaffected when the first three N-terminal residues were removed (IncE_2, IncE_3 and IncE_4) and gradually became weaker until IncE_7, after which binding was abolished. Results from IncE_6 are inconclusive due to the difficulty in successfully dissolving the peptides in buffer (n.d.). C-terminal truncations showed that IncE_14 and IncE_15 had similar high binding affinities to the reference, while the binding of IncE_16 and IncE_17 became progressively weaker and peptides shorter than IncE_17 showed no binding. This data indicates that the minimal IncE binding sequence retaining full SNX5 binding is GPAVQFFKGKNGSADQVILVT, and a shorter fragment VQFFKGKNGSADQVIL can bind to SNX5, albeit with a slightly reduced affinity. (**D**) Sequence alignment of IncE from different *C. trachomatis* serovars.

To determine the structure of the SNX5-IncE complex we generated a fusion protein encoding the human SNX5 PX domain (residues 22–170) and *C. trachomatis* IncE C-terminal sequence (residues 108–132) attached at the PX domain C-terminus *Figure 4—figure supplement 1A*. This construct readily crystallised in several crystal forms, and we were able to determine the structure of the complex in three different spacegroups (*Figure 4*; *Table 3*; *Figure 4—figure supplement 1B*). Confirming that the fusion does not alter complex formation, the short linker region is disordered, and the mode of IncE-binding to SNX5 is identical in all three structures (*Figure 4—figure supplement 1C and D*). Because of the higher resolution, we focus our discussions on the structure of the SNX5 PX-IncE complex observed in the $P2_12_12_1$ crystal form. The first three IncE N-terminal residues (Pro107, Ala108, Asn109) and the last three IncE C-terminal residues (Val130, Thr131, Gln132) were not modeled due to lack of electron density, suggesting disorder and matching precisely with our mapping experiments showing these residues are not necessary for SNX5 association.

The IncE sequence forms a long β-hairpin structure that binds within a complementary groove at the base of the extended α-helical insertion of the SNX5 PX domain and adjacent to the β-sheet sub-domain (*Figure 4A*; *Video 2*). The β-hairpin structure of IncE (N-terminal βA and C-terminal βB strands) is directly incorporated as a β-sheet augmentation of the β1, β2 and β3 strands of SNX5 (*Figure 4B*). The N-terminal βA strand of the IncE sequence (Gly111-Lys118) forms the primary interface with SNX5, making main-chain hydrogen bonds with the β1 strand of the SNX5 PX domain for the stable positioning of the IncE structure. The two anti-parallel β-strands of IncE are connected by a short loop (Gly119-Ala124) that makes no direct contact with the SNX5 protein, and the C-terminal IncE βB strand (Asp125-Val130) forms an interface with the extended α-helical region of the SNX5 PX domain.

Detailed views of the SNX5-IncE interface are shown in *Figure 4C, D and E*. Aside from main-chain hydrogen bonding to form the extended β-sheet, IncE engages in several critical side-chain interactions with the relatively hydrophobic SNX5 binding groove. At the N-terminus of the βA strand Val114 of IncE inserts into a pocket formed primarily by Tyr132 and Phe136 on the SNX5 α'' helix (*Figure 4C*). A major contribution comes from IncE Phe116, with π-stacking occurring with the Phe136 side-chain and hydrophobic docking with Val140 of SNX5 (*Figure 4D*). Adjacent to IncE Phe116 at the end of the βA strand Lys118 makes an electrostatic contact with SNX5 Glu144. Finally, at the C-terminal end of the IncE βB strand Val127 and Leu129 contact an extended SNX5 surface composed of Leu133, Tyr132 and Met106 (*Figure 4E*).

## Mutations in the SNX5-IncE interface disrupt complex formation in vitro and in cells

To verify the crystal structure we mutated residues from both SNX5 and IncE and measured their affinities using ITC (*Figure 5A and B*; *Table 2*). At the interface between SNX5 and IncE several side chains make key contributions to peptide recognition. Because Leu133 and Phe136 residues in SNX5 are located at the core of the IncE-binding interface, and also due to the structural rearrangements these residues make on binding (see below), we reasoned that L133D and F136A mutations would inhibit the interaction. Indeed these mutants abolished association with the IncE peptide (*Figure 5A*). The reciprocal mutations in IncE residues F116A and V127D also abolished binding to the SNX5 PX domain (*Figure 5B*), and the SNX6 and SNX32 PX domains (*Figure 5—figure supplement 1*), demonstrating the importance of these hydrophobic and π-stacking interactions for stable

**Table 2.** ITC data for SNX5 PX domain binding to truncated and mutated IncE peptides*.

| Protein | Peptide | Sequence | Kd (μM) | △H (kcal/mol) | T△S (kcal/mol) | △G (kcal/mol) | N |
|---|---|---|---|---|---|---|---|
| SNX5 PX | IncE_1 | PANGPAVQFFKGKNGSADQVILVTQ | 0.95 ± 0.07 | −6.9 ± 0.3 | −1.9 ± 0.05 | −8.2 ± 0.01 | 1.01 ± 0.01 |
| | IncE_2 | ANGPAVQFFKGKNGSADQVILVTQ | 1 | −5.0 | −2.6 | −8.1 | 0.98 |
| | IncE_3 | NGPAVQFFKGKNGSADQVILVTQ | 0.93 | −6.7 | −1.4 | −8.1 | 1.03 |
| | IncE_4 | GPAVQFFKGKNGSADQVILVTQ | 0.87 | −6.8 | −1.2 | −8.2 | 1.03 |
| | IncE_5 | PAVQFFKGKNGSADQVILVTQ | 2 | −5.9 | −1.2 | −8.3 | 0.99 |
| | IncE_6 | AVQFFKGKNGSADQVILVTQ | / | / | / | / | / |
| | IncE_7 | VQFFKGKNGSADQVILVTQ | 2.2 | −6.9 | −1.1 | −7.7 | 0.99 |
| | IncE_8 | QFFKGKNGSADQVILVTQ | No binding | / | / | / | / |
| | IncE_9 | FFKGKNGSADQVILVTQ | No binding | / | / | / | / |
| | IncE_10 | FKGKNGSADQVILVTQ | No binding | / | / | / | / |
| | IncE_11 | KGKNGSADQVILVTQ | No binding | / | / | / | / |
| | IncE_12 | GKNGSADQVILVTQ | No binding | / | / | / | / |
| | IncE_13 | KNGSADQVILVTQ | No binding | / | / | / | / |
| | IncE_14 | PANGPAVQFFKGKNGSADQVILVT | 0.72 | −5.1 | −1.6 | −8.4 | 1 |
| | IncE_15 | PANGPAVQFFKGKNGSADQVILV | 0.97 | −6.5 | −1.3 | −8.2 | 0.98 |
| | IncE_16 | PANGPAVQFFKGKNGSADQVIL | 1.1 | −5.6 | −1.4 | −8.12 | 0.99 |
| | IncE_17 | PANGPAVQFFKGKNGSADQVI | 8.7 | −2.7 | −2.5 | −6.9 | 0.99 |
| | IncE_18 | PANGPAVQFFKGKNGSADQV | No binding | / | / | / | / |
| | IncE_19 | PANGPAVQFFKGKNGSADQ | No binding | / | / | / | / |
| | IncE_20 | PANGPAVQFFKGKNGSAD | No binding | / | / | / | / |
| | IncE_21 | PANGPAVQFFKGKNGSA | No binding | / | / | / | / |
| | IncE_22 | PANGPAVQFFKGKNGS | No binding | / | / | / | / |
| | IncE_23 | PANGPAVQFFKGKNG | No binding | / | / | / | / |
| | IncE_24 | PANGPAVQFFKGKN | No binding | / | / | / | / |
| | IncE Q115A | PANGPAV**A**FFKGKNGSADQVILVTQ | 6.3 | −5.3 | −1.6 | −6.9 | 0.90 |
| | IncE F116D | PANGPAVQ**A**FKGKNGSADQVILVTQ | No binding | | | | |
| | IncE K118A | PANGPAVQFF**A**GKNGSADQVILVTQ | 2.8 | −6.0 | −1.5 | −7.5 | 0.91 |
| | IncE V127D | PANGPAVQFFKGKNGSADQ**D**ILVTQ | No binding | | | | |
| SNX5 PX L133D | IncE_1 | PANGPAVQFFKGKNGSADQVILVTQ | No binding | | | | |
| SNX5 PX F136A | IncE_1 | PANGPAVQFFKGKNGSADQVILVTQ | No binding | | | | |
| SNX5 PX E144A | IncE_1 | PANGPAVQFFKGKNGSADQVILVTQ | 15 | −9.9 | −3.1 | −13 | 0.99 |

*Except for IncE_1 all other peptide-binding experiments were performed only once.

complex formation. In contrast mutations predicted to disrupt an observed electrostatic contact (IncE K118A or SNX5 E144A) had little effect on binding. Thus the core hydrophobic interactions are critical for IncE binding but the peripheral electrostatic contact is not essential.

To confirm the role of IncE in direct SNX5 protein recruitment to the chlamydial inclusion we examined the localisation of GFP-tagged SNX5 in HeLa cells infected with *C. trachomatis* L2 (CTL2) for 24 hr (MOI ~0.5). Cells expressing the GFP-SNX5 protein showed clear and uniform recruitment to the limiting membrane of the inclusion as defined by mCherry-Rab25 (*Figure 6A*), which is consistent with the localisation observed by others (*Aeberhard et al., 2015*; *Mirrashidi et al., 2015*). In contrast, the GFP-SNX5 (F136A) mutant protein showed no recruitment to the chlamydial inclusion. The change in relative distribution of these GFP-SNX5 proteins on the inclusion was quantified for wildtype SNX5 (Mander's coefficient 0.67 ± 0.14) and GFP-SNX5 (F136A) (0.041 ± 0.051) (*Figure 6—*

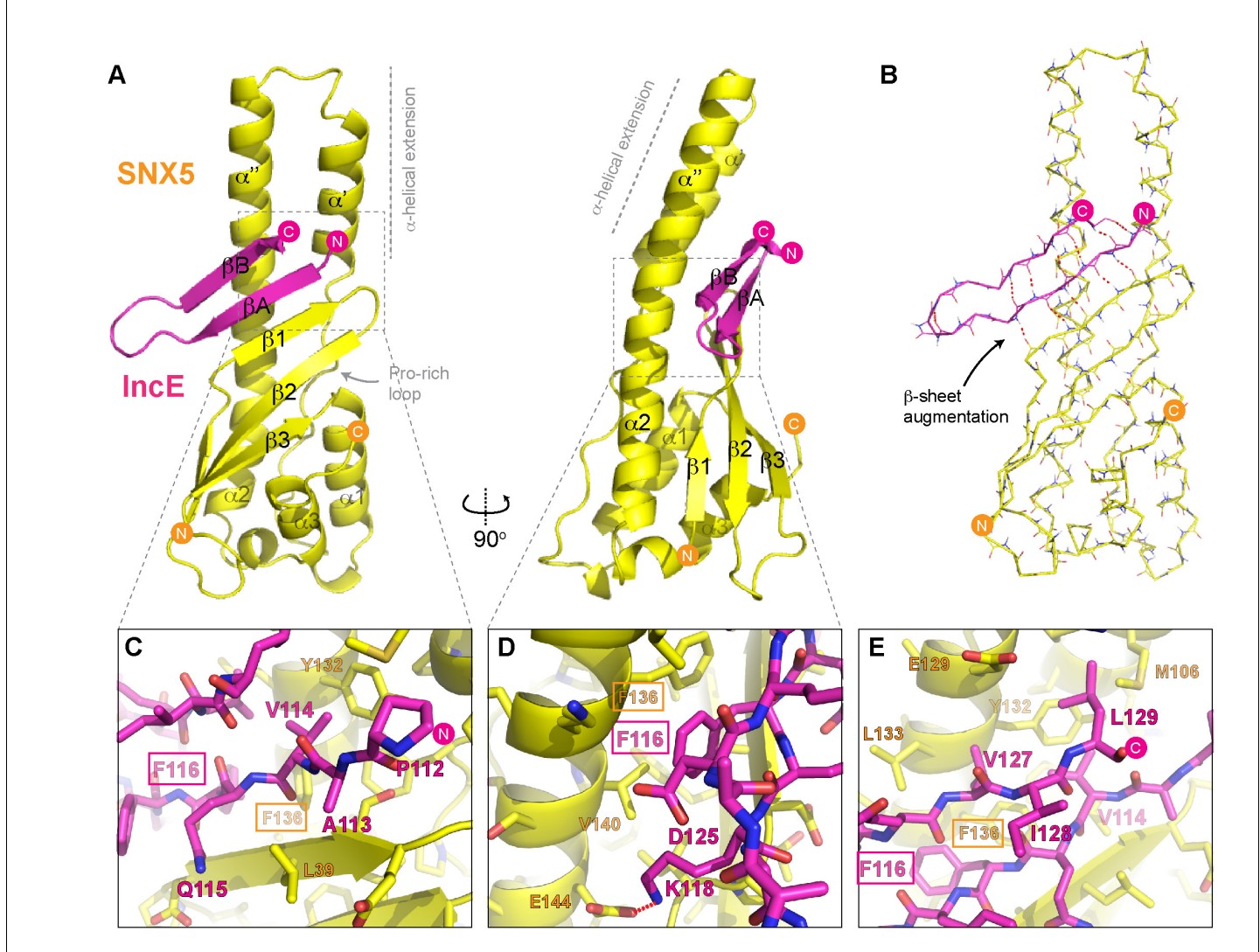

**Figure 4.** Structure of the SNX5 PX domain in complex with the IncE C-terminal domain. (**A**) Crystal structure of the SNX5 PX domain (yellow) in complex with IncE residues 107–132 (magenta) shown in cartoon representation. (**B**) Backbone atoms of the SNX5 and IncE proteins are shown to highlight the prominent β-sheet augmentation mediating the association between the two molecules. (**C**) Close up view of the SNX5-IncE interface highlighting specific contact areas at the N-terminus of the IncE peptide. (**D**) Close up of the SNX5-IncE interface highlighting specific contact areas at the hairpin loop of the IncE peptide shown at 90° to *Figure 4C*. (**E**). Close up of the SNX5-IncE interface highlighting contact areas at the C-terminus of the IncE peptide in approximately the same orientation as *Figure 4C*. Residues in SNX5 (Phe136) and IncE (Phe116) that are critical for binding based on mutagenesis are boxed.

The following figure supplement is available for figure 4:

**Figure supplement 1.** Supplementary images for SNX5-IncE crystal structures.

*figure supplement 1A*). Like wild-type GFP-SNX5 the GFP-SNX5 (F136A) mutant was recruited to punctate endosomal structures throughout the cytoplasm of these cells, and in addition was able to co-immunoprecipitate endogenous SNX1 in heterodimeric complexes identically to the wild-type GFP-SNX5 protein (*Figure 6—figure supplement 1B*). This implies that BAR-domain mediated heterodimer formation with SNX1 or SNX2 is required for endosomal recruitment, and is not perturbed by the IncE-binding mutation in the PX domain. Lastly, we tested the importance of IncE residues for SNX interaction in situ by expressing the GFP-tagged IncE C-terminal domain. The wild-type GFP-IncE(91-132) was recruited to endosomal structures via its interaction with SNX5-related proteins in

**Table 3.** Summary of crystallographic structure determination statistics*.

| Crystal | SNX5 PX-IncE Form 1 | SNX5 PX-IncE Form 2 | SNX5 PX-IncE Form 3 |
|---|---|---|---|
| PDB ID | 5TGI | 5TGJ | 5TGH |
| Data collection | | | |
| Wavelength (Å) | 0.95370 | 0.95370 | 0.95370 |
| Space group | $P2_12_12_1$ | $I2$ | $P3_2$ |
| Cell dimensions | | | |
| $a$, $b$, $c$ (Å) | 60.7, 67.5, 88.2 | 58.4, 80.3, 94.6 | 100.6, 100.6, 71.7 |
| $\alpha$, $\beta$, $\gamma$ (°) | 90, 90, 90 | 90, 97.2, 90 | 90, 90, 120 |
| Resolution (Å) | 60.7–1.98 (2.03–1.98) | 31.9–2.6 (2.72–2.60) | 50.3–2.80 (2.95–2.80) |
| $R_{merge}$ | 0.104 (0.525) | 0.153 (0.659) | 0.101 (0.713) |
| $R_{meas}$ | 0.112 (0.572) | 0.18 (0.777) | 0.124 (0.873) |
| $R_{pim}$ | 0.042 (0.225) | 0.096 (0.408) | 0.051 (0.363) |
| $<I>$ / $\sigma I$ | 12.4 (3.4) | 39.6 (3.2) | 11.7 (2.3) |
| Total number reflections | 178868 (11000) | 46691 (5757) | 115149 (16861) |
| Total unique reflections | 26075 (1805) | 13432 (1632) | 20001 (2923) |
| Completeness (%) | 100 (100) | 99.9 (100.0) | 100 (100) |
| Multiplicity | 6.9 (6.1) | 3.5 (3.5) | 5.8 (5.8) |
| Half-set correlation (CC(1/2)) | 0.997 (0.868) | 0.986 (0.55) | 0.997 (0.683) |
| Refinement | | | |
| Resolution (Å) | 45.1–1.98 (2.02–1.98) | 31.9–2.6 (2.69–2.60) | 41.2–2.8 (2.87–2.80) |
| No. reflections/No. $R_{free}$ | 26021/2000 | 13421/1342 (1208/134) | 19975/1972 (1301/144) |
| $R_{work}$/$R_{free}$ | 0.192/0.214 (0.221/0.246) | 0.199/0.242 (0.276/0.332) | 0.236/0.254 (0.329/0.372) |
| No. atoms | | | |
| Protein | 2579 | 2619 | 5189 |
| Solvent | 281 | 69 | 0 |
| Average $B$-factor (Å$^2$) | 31.8 | 42.5 | 56.0 |
| R.m.s deviations | | | |
| Bond lengths (Å) | 0.012 | 0.011 | 0.015 |
| Bond angles (°) | 1.27 | 1.15 | 1.27 |

*Highest resolution shell is shown in parentheses.

both uninfected and infected HeLa cells (*Figure 6B*; *Figure 6—figure supplement 1C*). In contrast however, GFP-IncE(91-132)(F116D), a SNX5-binding mutant, was exclusively cytosolic. Note that neither IncE construct is localised to the inclusion, as expected due to lack of signal peptides and transmembrane domains (*Figure 6—figure supplement 1C*).

## A model for SNX-BAR recruitment to the inclusion membrane by IncE

Superposition of the SNX5-IncE complex with the SNX5 PX domain in the apo state (*Koharudin et al., 2009*) reveals a significant conformational change in the α-helical extension, as well as localized alterations in the loop between the β1 and β2 strands to accommodate peptide binding (*Figure 7A*). In essence the IncE β-hairpin acts as a tether between the core PX fold and extended α-helical hairpin, pulling the two sub-structures closer together. Overall the α-helical extension undergoes a maximal movement of ~8–10 Å at the furthest tip, facilitated by the flexibility of the structure following the Pro-rich linker and an apparent hinge-point at Pro97 (*Figure 7A* upper panel). In the immediate vicinity of Pro97 the SNX5 loop that encompasses Asp43 is significantly

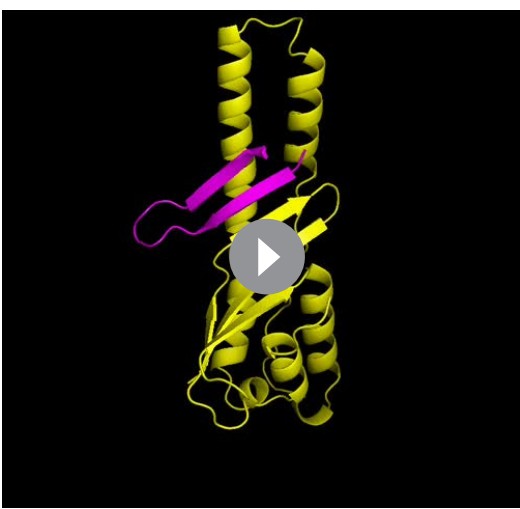

shifted and stabilized by the repositioning of Arg103. At both the start of the first α' helix of the extension and the end of the second α'' helix more subtle changes occur in the positions of Met106, Leu128, Tyr132, Leu133 and Phe136. These changes result in formation of the hydrophobic pocket that engages the IncE side-chains Val114, Phe116, Val127 and Leu129 (*Figure 7A*, middle and lower panels).

To better understand how IncE can recruit the SNX5-containing SNX-BAR complexes to inclusion membranes we constructed an in silico model of the SNX5-SNX1 heterodimeric PX-BAR proteins (*Figure 7B*). Consistent with the length of the IncE C-terminal cytoplasmic sequence the model predicts that the IncE sequence will bind to the surface of SNX5 close to, but oriented away from, the inclusion membrane. While PX domains are commonly able to recognise PtdIns3*P* lipid headgroups, SNX5-related proteins lack the typical binding pocket (*Figure 7B* right panel), and there is some controversy regarding their ability to mediate specific membrane interactions (*Koharudin et al., 2009*;

**Video 2.** Animation highlighting the mechanism of interaction between SNX5 and IncE. The SNX5 PX domain is shown in yellow ribbons and the IncE peptide is shown in magenta.

*Teasdale and Collins, 2012*). We propose that in the context of *C. trachomatis* infection, SNX5-related proteins are directly associated with the inclusion via IncE protein-protein interactions in a

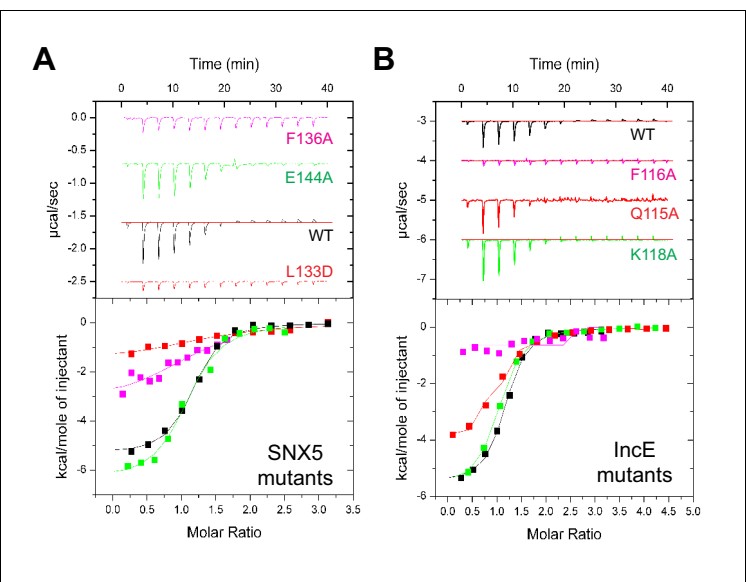

**Figure 5.** Mutations in the SNX5 and IncE proteins prevent complex formation in vitro. (**A**) ITC experiments testing the effect of SNX5 mutations on IncE binding. Both L133D and F136A mutations prevented IncE binding, but the A144A mutation had little effect. (**B**) ITC experiments testing the effect of IncE mutations on SNX5 binding. Both F116A and V127D blocked SNX5 interaction, while Q115A had a partial effect and K118A had no effect on association.
The following figure supplement is available for figure 5:

**Figure supplement 1.** SNX6 and SNX32 PX domains bind IncE at the same site as SNX5.

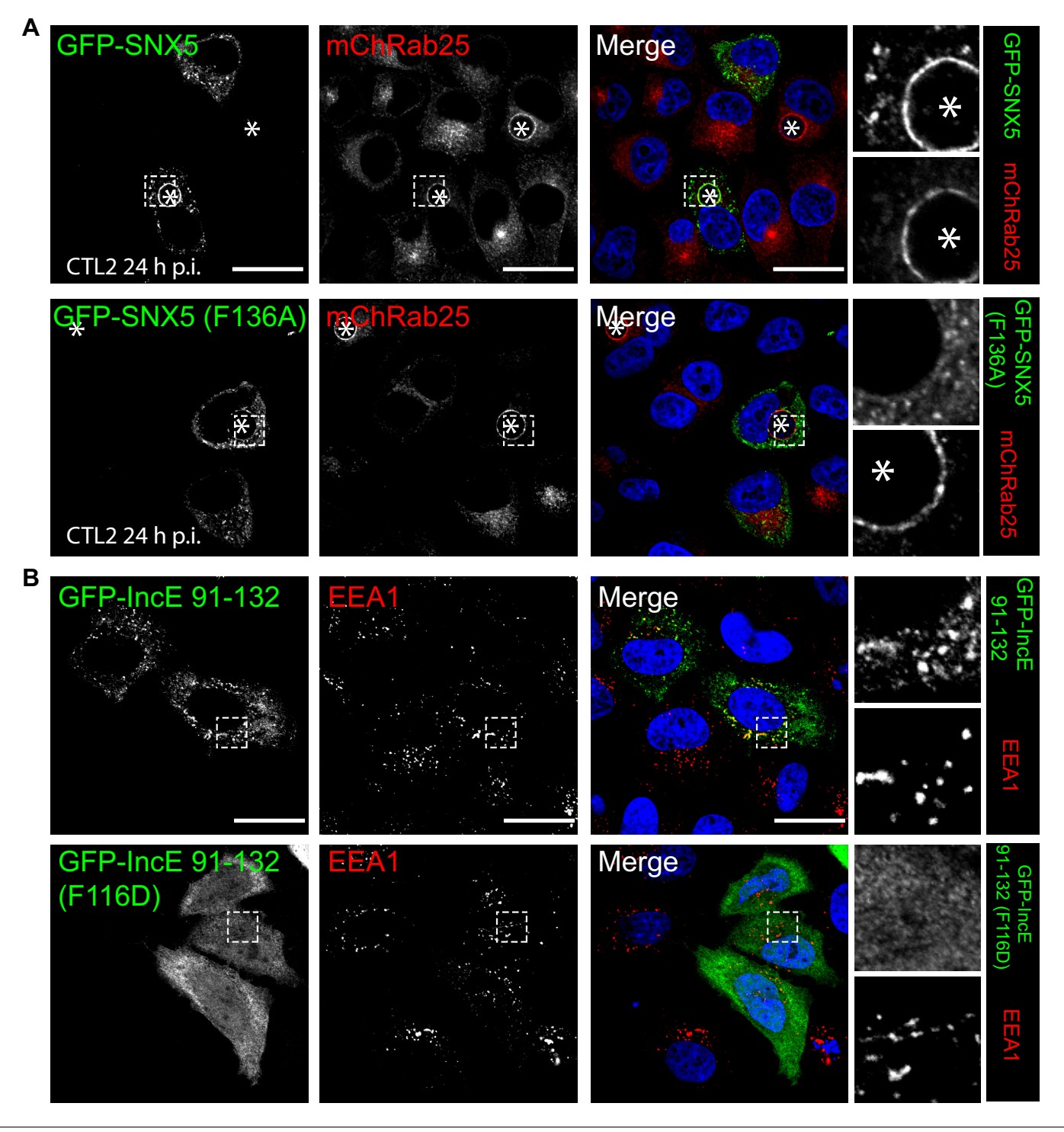

**Figure 6.** Mutations in the SNX5 and IncE proteins prevents their interaction in cells. (**A**) Single amino-acid mutation in the PX domain of the SNX5 (F136A) abolishes recruitment to the chlamydial inclusion. HeLa cells stably expressing mCherry-Rab25 (red) were transfected transiently with GFP-SNX5 or GFP-SNX5 (F136A) (green) and infected with *Chlamydia trachomatis* L2 for 18–24 hr. The cells were fixed and the nucleic materials were counter-stained with DAPI (blue). (**B**) HeLa cells were transfected transiently with GFP-IncE(91-132) or GFP-IncE(91-132)(F116D) (green) and co-labelled for the early endosomal marker EEA1 (red). Mutation in the SNX5 binding IncE peptide (F116D) abolishes recruitment to endosomal structures. *Represents the inclusion. Scale bar 20 μm.

The following figure supplement is available for figure 6:

*Figure 6 continued on next page*

*Figure 6 continued*

**Figure supplement 1.** GFP-IncE C-terminal domain is localised to endosomes but not inclusions.

phosphoinositide-independent manner, and are able to recruit their heterodimeric partners SNX1 and SNX2 (*Sierecki et al., 2014*; *van Weering et al., 2012*; *Wassmer et al., 2009*). The PX-BAR-domain containing complexes are then localised to the inclusion in a retromer-independent manner (*Mirrashidi et al., 2015*), and may contribute to the formation of the dynamic inclusion-associated membrane tubules.

Interestingly, when a cross-species evolutionary analysis of side-chain conservation in the SNX5-related proteins is performed it is clear that the IncE peptide binds a hydrophobic surface groove that is strictly conserved in this protein family (*Figure 7C*). This very strongly implies that the site is normally engaged in a protein-protein interaction with an as yet unidentified binding partner(s) required for SNX5's regular biological function, and that IncE is directly competing for this interface.

## Discussion

Although more than fifty putative Incs have been identified in *C. trachomatis*, the exact roles of these inclusion membrane proteins are still poorly understood. *Chlamydiae* manipulate the host cellular and signaling networks via interactions between the cytoplasmic region of Incs and numerous host cell proteins. Recent studies reported retrograde trafficking proteins as significant components of the inclusion, with sorting nexin family members being particularly enriched (*Aeberhard et al., 2015*; *Mirrashidi et al., 2015*). In this study, we present the first reported crystal structure of a chlamydial inclusion protein (IncE) binding to its host effector protein (SNX5). While the detailed mechanism of IncE-mediated protein recruitment will be specific to this family member, the principle of extended cytoplasmic Inc sequences engaging with cellular host proteins on the inclusion is certain to be a general one. A simple analogy would be to consider the Inc proteins as being like a molecular 'velcro' that recognises and attaches host machinery needed for bacterial replication and survival.

The manipulation of endocytic transport machinery is clearly critical for the obligate intracellular survival of *C. trachomatis* (*Aeberhard et al., 2015*; *Mirrashidi et al., 2015*; *Moore and Ouellette, 2014*). In addition to *C. trachomatis,* SNX1, SNX2, SNX5, SNX6 and the associated retromer complex have also been directly implicated in the cellular pathogenesis of *Coxiella burnetii* (*McDonough et al., 2013*), *Salmonella* enterica serovar *Typhimurium* (*Bujny et al., 2008*), hepatitis C virus (*Yin et al., 2016*), human papilloma virus (*Ganti et al., 2016*; *Popa et al., 2015*), and *Legionella pneumophila* (*Finsel et al., 2013*). Broadly then the manipulation of SNX proteins and endosomal trafficking machinery by viral and bacterial pathogens is a common occurrence during intracellular infection, and points to a wide-ranging role in host-pathogen interactions.

Typically PX domains of sorting nexins, including SNX1 and SNX2 (*Cozier et al., 2002*; *Zhong et al., 2005*), play an important role in endosomal membrane recruitment by binding the endosome-enriched lipid PtdIns3$P$ through four conserved residues (*Mas et al., 2014*; *Teasdale and Collins, 2012*). These residues are conserved in most PX domains including in SNX1 and SNX2, but are entirely absent in SNX5, SNX6 and SNX32. Although there is evidence for the weak association of the SNX5 PX domain with the lipid PtdIns(4,5)$P_2$ from nuclear magnetic resonance (NMR) spectroscopy experiments (*Koharudin et al., 2009*), the crystal structure does not point to a clear binding mechanism. A second feature that sets SNX5-related proteins apart from the rest of the SNX family is the presence of an extended α-helical insertion. Our work confirms the central importance of this unique insert for the binding of the IncE inclusion protein, and provides the first clear description of how a PX domain can function as a protein-protein interaction scaffold as opposed to a lipid-binding domain.

The high degree of conservation in the IncE binding surface of SNX5 implies that this site is critical for the normal function of SNX5 and its homologs. Previously, the expression of a GFP-tagged IncE C-terminal domain was shown to interfere with the SNX5/SNX6-dependent retrograde trafficking of the cation-independent mannose-6-phosphate receptor (CI-MPR) (*Mirrashidi et al., 2015*). Combined with our structural data, this infers that IncE is mimicking and interfering with SNX5/SNX6-mediated protein interactions, with a ligand(s) required for normal endosomal trafficking that

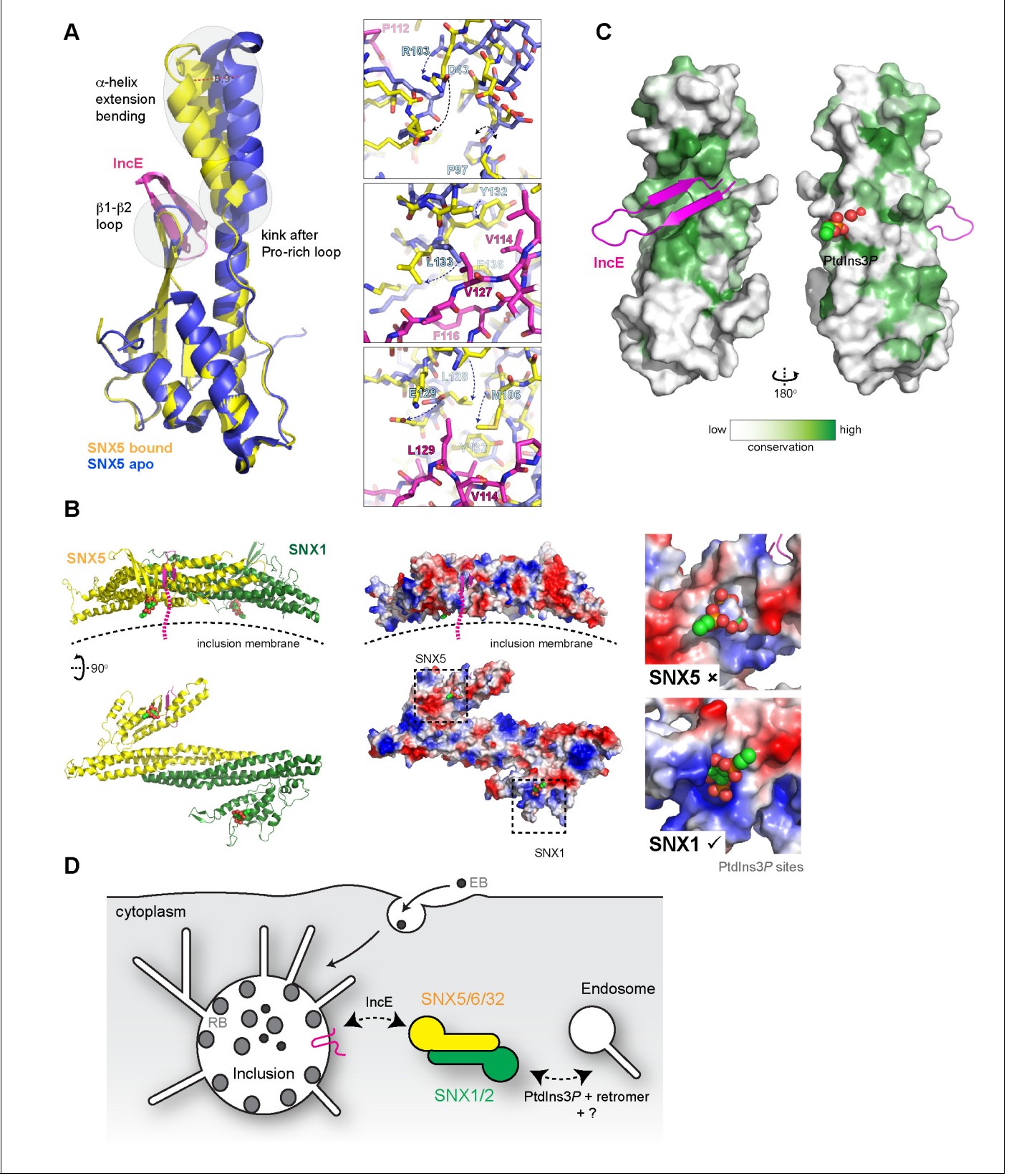

**Figure 7.** Conformational changes in SNX5 and a model for SNX-BAR recruitment to inclusion membranes. (**A**) Comparison of the SNX5-IncE complex (yellow-magenta) with the previously reported apo- SNX5 PX domain crystal structure (blue) (PDB ID 3HPB)(**Koharudin et al., 2009**). The α-helical extension undergoes a significant displacement in the bound state. The enlarged panels to the right show several close-up views of the binding pocket

*Figure 7 continued on next page*

*Figure 7 continued*

highlighting conformational changes that are required to accommodate IncE. (**B**) A model for the SNX5-SNX1 PX-BAR heterodimer and its interaction with IncE at the inclusion membrane. The PX-BAR structure was modeled in silico (see methods). The left panel shows cartoon representations of the structure, viewed from the side and from the membrane surface. Middle panels show the same structures in electrostatic surface representation (red, negative; blue positive). The right panels show close ups of the putative PtdIns3*P*-binding pocket in SNX1 and SNX5, with a PtdIns3*P* head-group (shown in spheres) docked by aligning the previous SNX9 crystal structure (*Pylypenko et al., 2007*). SNX1 has a canonical PtdIns3*P* pocket, while SNX5 lacks a clear site for lipid head-group binding. (**C**) Sequence conservation of SNX5-related proteins was calculated and plotted using CONSURF. The surface representation indicates exposed side-chains that are evolutionarily conserved in green. The IncE peptide binds to a highly conserved surface groove, while the putative phosphoinositide binding region (*Koharudin et al., 2009*) on the opposite face is neither highly conserved nor poised to allow docking. (**D**) Cartoon model depicting the recruitment of SNX5 and related proteins to the inclusion membrane. Heterodimers with SNX1 or SNX2 will be recruited via IncE in infected cells, and this recruitment will be in competition with the binding of SNX1 and SNX2 to PtdIns3*P* for normal endosomal association, as well as interactions with other proteins including retromer and unidentified molecules that potentially bind to the conserved groove of the SNX5 PX domain.

remains to be discovered. Once recruited to the inclusion, SNX-BAR proteins are localized to the bulk membrane and dynamic tubules. While it is logical to imagine they could play a positive role in the sculpting of the inclusion, this is somewhat difficult to reconcile with the effect of SNX5 and SNX6 knockdown, which results in an increased production of *C. trachomatis* infectious progeny. Alternatively, although a pool of SNX5/SNX6 and associated SNX1/SNX2 proteins remain on endosomes in *C. trachomatis* infected cells, their sequestering by the chlamydial inclusion may interfere with normal endosomal trafficking (*Figure 7D*). It was thus proposed that the role of IncE could be to compete for SNX-retromer endosomal interactions, resulting in the breakdown of normal trafficking of the CI-MPR and lysosomal hydrolases and hence perturbation of the endolysosomal system's capacity for bacterial destruction (*Aeberhard et al., 2015*; *Mirrashidi et al., 2015*). Defining the precise role of SNX proteins and other endocytic machinery in chlamydial infection will clearly require further study.

In conclusion, our work provides novel molecular insights into the mechanism of SNX protein coercion by the IncE chlamydial effector, and presents a blueprint for future studies of other inclusion protein activities. In addition, our results provide a possible clue to understanding how SNX5-related molecules mediate protein interactions required for canonical cell trafficking pathways.

## Materials and methods

### Peptides

All synthetic peptides used for isothermal titration were purchased from Genscript (USA). For ITC experiments, peptides were weighed and dissolved in 50 mM Tris (pH 8.0) and 100 mM NaCl (ITC buffer) to make a stock peptide concentration of 2 mM, which was diluted to 0.75 mM before use.

### Antibodies and reagents

Polyclonal antibodies against *C. trachomatis* HtrA were generated previously (*Huston et al., 2008*). Monoclonal antibodies against EEA1 (610457, 1:100), SNX1 (611483,1:200) and SNX2 (611308, 1:200) were supplied by BD Bioscience. Monoclonal antibodies against the myc epitope (9B11, 1:2000) were supplied by Abcam. Rabbit polyclonal antibodies against GFP (A-6455, 1:500) were purchased from Molecular Probes (Invitrogen). Rabbit polyclonal antibodies against Rab5 (C8B1, 1:100) were from Cell Signaling Technology. Goat polyclonal antibodies against Vps35 (IMG-3575, 1:400) were from Imgenex. Secondary antibodies were purchased from Molecular Probes (Life Technologies) and Li-Cor Bioscience. Wortmannin was supplied by Sigma-Aldrich (W1628). VPS34-In1 was from Merck Millipore (532628).

### Molecular biology and expression constructs

The IncE sequence used in this study is from the L3 serovar L3/404/LN (NCBI reference WP_015506602) (*Harris et al., 2012*). The pGEX-4T-2 bacterial expression plasmid encoding the human SNX5 PX domain (residues 22–170) was generated using a standard PCR-based cloning strategy, and its identity confirmed by sequencing. All other bacterial expression constructs for human

SNX proteins were synthesized and cloned into pGEX-4T-2 by Genscript (USA). These included the SNX5 PX domain IncE fusion (SNX5 residues 22–170 with IncE residues 108–132 fused at the C-terminus (*Figure 4—figure supplement 1A*), SNX6 (residues 29–170), SNX32 (residues 17–166), and SNX5 PX domain mutants. The pcDNA3.1-N-eGFP mammalian expression constructs encoding full-length human SNX5, SNX5(F136A), IncE(91-132) and IncE(91-132)(F116D) with N-terminal GFP-tags were generated by Genscript (USA). The pCMU-myc-SNX5 was as described previously (*Kerr et al., 2006*), and the SNX6 and SNX32 genes cloned into the pcDNA3.1-nMyc vector at BamHI and XhoI restriction sites (*Kerr et al., 2012*). SNX5, SNX32 and SNX8 were also cloned by polymerase chain reaction, restriction digest and ligation into pEGFP-C1 for expression with N-terminal GFP tags as described previously (*Wang et al., 2010*).

## Recombinant protein expression and purification

All proteins except SNX5 PX domain mutants were expressed in *Escherichia coli* Rosetta cells, whereas mutant constructs were expressed in BL21 Codon Plus supplemented with appropriate antibiotics. Single colonies from cultures grown on LB agar plates were inoculated into 50 mL LB$^{2+}$ with ampicillin (0.1 mg/mL) and chloramphenicol (0.1 mg/mL), and grown at 37°C with shaking overnight. The following day, 30 mL from the overnight culture was used to inoculate 1 L LB media containing ampicillin (0.1 mg/mL) and chloramphenicol (0.1 mg/mL) and incubated at 37°C. Cells were grown to an optical density (OD) of 0.5–0.6 at 600 nm and induced with 0.5 mM isopropyl-$\beta$-D-thiogalactopyranoside (IPTG) (except for the SNX5-IncE fusion, where expression was induced at OD$_{600}$ of 0.8 with 1 mM IPTG). Cultures were incubated with shaking overnight at 18°C until the cells reach an O.D of 3.0 (~24 hr). Cells were harvested using a Beckman rotor JLA 8.1000 at 4000 RPM for 30 min at 4°C. Pellets were resuspended in 10 mL lysis buffer (50 mM Tris (pH 8.0), 100 mM NaCl, 5% glycerol, 1 mM DTT, 0.1 mg/ml benzamidine, 0.1 mg/ml DNase) per litre of culture. The cells were subjected to cell disruption and centrifugation at 18,000 RPM for 30 min at 4°C. The soluble fractions were first purified using affinity chromatography with glutathione-sepharose, and when required the GST tags were cleaved by thrombin while still bound to the column. The proteins were eluted in 50 mM Tris (pH 8.0), 100 mM NaCl, 5% glycerol, and 1 mM DTT, and then further polished using gel filtration chromatography (Superdex 200, GE healthcare) in a buffer containing 50 mM Tris (pH 8.0), 100 mM NaCl. The fractions corresponding to the respective proteins were then pooled and used directly for ITC or were further concentrated for crystallization.

## Isothermal titration calorimetry

ITC experiments were performed on a Microcal iTC200 instrument at 25°C. The proteins were buffer exchanged into ITC buffer (50 mM Tris (pH 8.0) and 100 mM NaCl) by gel filtration prior to ITC experiments. IncE peptides at 750 µM were titrated into 50 µM PX domain samples. The binding data was processed using ORIGIN 7.0 with a single site binding model to determine the stoichiometry (n), the equilibrium association constant $K_a$ ($1/K_d$), and the enthalpy ($\triangle H$). The Gibbs free energy ($\triangle G$) was calculated using the equation $\triangle G = -RT\ln(K_a)$; binding entropy ($\triangle S$) was calculated by $\triangle G = \triangle H - T\triangle S$. Three experiments were performed for each set of samples to determine the average ± standard error of the mean (SEM) for thermodynamic quantities, except for the peptide truncation experiments where only single experiments were performed. For these truncated peptide experiments, all experiments were performed using the same batch of protein to allow direct comparions to be made.

## Crystallization, data collection and structure determination

The SNX5 PX domain fusion with IncE was concentrated to 15 mg/ml for crystallization. Eight 96-well crystallization hanging-drop screens were set up using a Mosquito Liquid Handling robot (TTP LabTech) at 20°C. Optimized diffraction-quality crystals were obtained using streak seeding in sitting drop vapor diffusion plates. The crystallisation solution for crystal form 1 was 0.2 M KSCN, 25% PEG 2K MME, 100 mM sodium acetate (pH 5.5), for crystal form 2 was 0.1 M NaCl, 0.1 M MgCl2, 0.1 M Nacitrate (pH 3.5), 12 % PEG 4000, and for crystal form 3 was 1.26 M (NH4)2SO4, acetate (pH 4.5), 0.2 M NaCl. Data were collected at the Australian Synchrotron MX1 and MX2 Beamlines, integrated with iMOSFLM (*Battye et al., 2011*), and scaled with AIMLESS (*Evans and Murshudov, 2013*) in the CCP4 suite (*Winn et al., 2011*). The structures were initially solved by molecular replacement with

PHASER (*McCoy et al., 2007*) using the apo-SNX5 PX domain crystal structure as the input model (PDB code 3HPB), minus the extended α-helical domain. The resulting model was rebuilt with COOT (*Emsley et al., 2010*), followed by repeated rounds of refinement with PHENIX (*Adams et al., 2011*). All structural figures were generated using PyMOL (DeLano scientific).

## Cell culture and transfections

HeLa cells stably expressing mCherry-Rab25 were previously generated within the lab (*Teo et al., 2016*) and were maintained in DMEM (Gibco) supplemented with 10% (v/v) FCS (Bovogen) and 2 mM L-glutamine (Invitrogen) in a humidified air/atmosphere (5% $CO_2$) at 37°C. Cells were transfected at 70% confluence with pcDNA3.1-N-eGFP plasmid constructs using Lipofectamine 2000 as per manufacturer's protocol (Invitrogen) and examined 18–24 hr later. The HeLa cell line used in this study was from America Type Culture Collection (#ATCC CCL2). Parental and stable cells lines were negative for mycoplasma by DAPI staining, and authenticated by STR profiling (Cell Bank Australia). For inhibitor treatments, cells were treated with either 100 nM wortmannin or 1 μM Vps34-IN1 for 1 hr.

## Chlamydial infection assays

C.C. trachomatis serovar L2 (ATCC VR-902B) was used to infect cells at a multiplicity of infection (MOI) of ~0.5. Cells were infected 2 hr post-transfection in normal DMEM (Gibco) supplemented with 10% (v/v) FCS (Bovogen) and 2 mM L-glutamine (Invitrogen) in a humidified air/atmosphere (5% $CO_2$) incubator at 37°C. After 2 hr media was replaced with fresh media.

## Microscopy

Transfected and infected cells (18–24 hr post-infection) were fixed with 4% paraformaldehyde, permeabilised using TritonX-100 (Sigma) and immunolabeled as described previously (*Teo et al., 2016*) and counter-stained with DAPI. The coverslips were imaged using a confocal laser-scanning microscope (LSM 710 meta, Zeiss) with 63x oil immersion objective. Time-lapse videomicroscopy was carried out on individual live cells using a Nikon Ti-E inverted deconvolution microscope using a 40x, 0.9 Plan Apo DIC objective, a Hamamatsu Flash 4.0 4Mp sCMOS monochrome camera and 37°C incubated chamber with 5% $CO_2$. GFP was excited with a 485/20 nm LED and captured using a 525/30 nm emission filter, and mCherry was excited using a 560/25 nm LED and captured using a 607/36 nm emission filter. Data was processed using ImageJ (https://imagej.nih.gov/ij/) and compiled using Adobe Illustrator CS6.

## Image quantification

The immunofluorescence colocalisation of GFP-SNX5 with chlamydial inclusion membranes (*Figure 6A*; *Figure 6—figure supplement 1A*) imaged on a confocal microscope was measured by Mander's correlation coefficient of red pixel (EEA1 or mCherry-Rab25) over green pixel (GFP-SNX5) signals, which were determined using ImageJ (https://imagej.nih.gov/ij/) with the JACoP plugin (*Bolte and Cordelières, 2006*). Punctate structures were automatically counted using ImageJ analyse particle tool across total of 10 cells from two biological replicates. To quantify the effect of PI3K inhibitors on SNX recruitment (*Figure 1—figure supplement 2*), Z-stacks were captured with a Zeiss 710 confocal laser scanning microscope using a 40x objective. Maximum projections were generated with FIJI (https://fiji.sc/) and Pearson's correlation coefficients for individual cells determined using the FIJI 'Coloc 2' function with Costes threshold regression and 100 Costes randomisations. Colocalization analyses were conducted on two independent experiments from five images per condition each containing at least 20 cells (>100 cells analysed per condition).

## Co-precipitation of GFP-SNX5 and endogenous SNX1

HeLa cells were transfected with pcDNA3.1-N-eGFP plasmid constructs overnight at 70% confluence and the cells were lysed using lysis buffer ($H_2O$, 50 mM HEPES, 150 mM NaCl, 1% Triton-X100, 10 mM $Na_4P_2O_7$, 30 mM NaF, 2 mM $Na_3VO_4$, 10 mM EDTA, 0.5 mM AEBSF and protease inhibitor cocktail). Cell lysates were incubated with GFP nano-trap agarose beads (Protein Expression Facility, UQ) after preclear using protein G-agarose beads (Invitrogen). Protein complexes attached to the beads were detached by boiling for 5 min with 5x denaturing and reducing buffer (0.625 M Tris pH

6.8, 50% glycerol, 10% SDS, 0.25% Bromophenol blue and 500 mM DTT). Denatured and reduced proteins were separated by molecular mass using SDS-PAGE. Proteins were transferred onto PVDF-FL membrane (Immobilon) and were detected by immunoblotting with polyclonal anti-GFP and monoclonal SNX1 antibodies, and near-infrared fluorescent dyes (LI-COR). Immunolabelled proteins were visualised using LI-COR Odyssey imaging system.

## Modelling of the SNX5-SNX1 heterodimer

Human SNX5 and SNX1 sequences were submitted to the PHYRE2 server for automated homology-based model building (*Kelley et al., 2015*). For both proteins the top scoring modelling template was the crystal structure of the SNX9 PX-BAR domains (PDB ID 2RAJ) (*Pylypenko et al., 2007*) with Confidence Scores of 100% (and sequence identities of 19% and 16% respectively). The PX domain of the SNX5 model generated using this structural template was missing the extended α-helical insert, so to complete the model the SNX5 PX domain-IncE complex was substituted and a dimer of SNX5 and SNX1 PX-BAR domains generated by overlaying with the SNX9 dimer in the PtdIns3*P*-bound state (PDB ID 2RAK) (*Pylypenko et al., 2007*). The resulting model was subjected to simple energy minimisation in PHENIX (*Adams et al., 2011*). Conservation of surface residues was computed using the CONSURF server (*Ashkenazy et al., 2016*).

## Data deposition

Structural data are deposited in the protein data bank (PDB) under accession numbers 5TGI, 5TGJ, and 5TGH. Raw diffraction images are available on the University of Queensland eSPACE server (http://espace.library.uq.edu.au/view/UQ:409277).

## Acknowledgements

The authors would like to acknowledge support from the staff and facilities of the University of Queensland Remote Operation Crystallization and X-ray (UQ ROCX) facility, and the Australian Synchrotron. Microscopy was performed at the Australian Cancer Research Foundation (ACRF)/Institute for Molecular Bioscience (IMB) Dynamic Imaging Facility for Cancer Biology. Elements of this research utilised equipment and support provided by the QLD node of the National Biologics Facility (www.nationalbiologicsfacility.com), an initiative of the Australian Government being conducted as part of the NCRIS National Research Infrastructure for Australia. This work is supported by the Australian Research Council (ARC) (DP0985029; DP150100364) and National Health and Medical Research Council (NHMRC) (APP1058734; 606788). RDT is supported by an NHMRC Senior Research Fellowship (APP1041929), MCK is supported by an Australian Research Council Discovering Early Career Researcher Award (DE120102321), and BMC is supported by an NHMRC Career Development Fellowship (APP1061574).

## Additional information

### Funding

| Funder | Grant reference number | Author |
| --- | --- | --- |
| National Health and Medical Research Council | APP1058734 | Brett M Collins |
| National Health and Medical Research Council | 606788 | Rohan D Teasdale |
| National Health and Medical Research Council | APP1041929 | Brett M Collins |
| National Health and Medical Research Council | APP1061574 | Rohan D Teasdale |
| Australian Research Council | DP0985029 | Brett M Collins |
| Australian Research Council | DP150100364 | Brett M Collins |
| Australian Research Council | DE120102321 | Markus C Kerr |

The funders had no role in study design, data collection and interpretation, or the decision to submit the work for publication.

## Author contributions

BP, Conceptualization, Validation, Investigation, Visualization, Methodology, Writing—original draft, Writing—review and editing; HSK, Data curation, Validation, Investigation, Visualization, Writing—original draft, Writing—review and editing; MCK, Data curation, Formal analysis, Supervision, Validation, Investigation, Visualization, Methodology, Writing—original draft, Writing—review and editing; WMH, Data curation, Validation, Investigation, Writing—original draft, Writing—review and editing; RDT, Conceptualization, Resources, Data curation, Formal analysis, Supervision, Validation, Investigation, Visualization, Methodology, Writing—original draft, Project administration, Writing—review and editing; BMC, Conceptualization, Resources, Formal analysis, Supervision, Funding acquisition, Investigation, Visualization, Writing—original draft, Project administration, Writing—review and editing

## Author ORCIDs

Brett M Collins, http://orcid.org/0000-0002-6070-3774

## Additional files

### Major datasets

The following datasets were generated:

| Author(s) | Year | Dataset title | Dataset URL | Database, license, and accessibility information |
| --- | --- | --- | --- | --- |
| Collins B | 2016 | Structure of the SNX5 PX domain | http://espace.library.uq.edu.au/view/UQ:409277 | Publicly available at the University of Queensland eSpace (UQ: 409277). |
| Collins B, Paul B | 2017 | Structure of the SNX5 PX domain in complex with chlamydial protein IncE in space group P212121 | http://www.rcsb.org/pdb/explore/explore.do?structureId=5TGI | Publicly available at the RCSB Protein Data Bank (accession no. 5TGI) |
| Collins B, Paul B | 2017 | Structure of the SNX5 PX domain in complex with chlamydial protein IncE in space group I2 | http://www.rcsb.org/pdb/explore/explore.do?structureId=5TGJ | Publicly available at the RCSB Protein Data Bank (accession no. 5TGJ) |
| Collins B, Paul B | 2017 | Structure of the SNX5 PX domain in complex with chlamydial protein IncE in space group P32 | http://www.rcsb.org/pdb/explore/explore.do?structureId=5TGH | Publicly available at the RCSB Protein Data Bank (accession no. 5TGH) |

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
