## [Decision Letter]

Thank you for submitting your article "Structural basis for the hijacking of endosomal sorting nexin proteins by *Chlamydia trachomatis*" for consideration by *eLife*. You will be pleased to learn that your article has been favorably reviewed by three peer reviewers, one of whom, Suzanna Pfeffer, is a member of our Board of Reviewing Editors, and the evaluation has been overseen by Vivek Malhotra as the Senior Editor. The other following individuals involved in review of your submission have agreed to reveal their identity: Oliver Daumke and Volker Haucke.

The reviewers have discussed the reviews with one another and the Reviewing Editor has drafted this decision to help you prepare a revised submission.

Using quantitative proteomics, Heuer and colleagues previously found that SNX 5 and certain other SNX proteins are enriched on *Chlamydia* inclusions; in particular, SNX5 controls the *C. trachomatis* infection and retrograde trafficking is essential for infectious progeny formation. Engel and colleagues reported similar findings and more specifically, that the C-terminal portion of IncE binds to sorting nexins (SNXs) 5/6 of the retromer. Both reports showed that depletion of retromer components enhances progeny production, indicating that *Chlamydia* may hijack retromer to avoid degradative trafficking.

This manuscript reports the crystal structure of IncE C-terminal residues in complex with the PX domain of SNX5; in addition, it presents a protein binding interface for a PX domain in a subfamily of PX domain proteins that does not contain a functional phosphoinositide binding domain. Comparable binding affinity is seen for SNX 5, 6 and 32 PX domains but not that of SNX1. The story is compact and of high quality; it is an interesting variation on a theme of Chlamydial hijacking and the reviewers support presentation in *eLife* because it highlights an interesting biological interaction of medical importance: the elucidation of the molecular basis for hijacking of endosomal SNX5/6 and possibly 32 by bacterial IncE is important and of interest to the readers of *eLife*. Overall, the data support the main conclusions of the paper, however a few aspects should be strengthened by a few straightforward, additional experiments.

1) The authors provide qualitative data (e.g. Figure 4) from overexpression experiments in *Chlamydia* infected HeLa cells as an assay for IncE-mediated GFP-SNX5 recruitment. The staining in Figure 4 is hard to see and quantitative image analysis from multiple experiments will greatly strengthen the present story. The presence of endogenous endosomal SNX proteins that may interfere with recruitment and result in additional effects (e.g. as they may heterodimerize) that make the results difficult to interpret. Thus it would be worth testing IncE-mediated recruitment of SNX5 to bacterial inclusions using quantitative image analysis in HeLa cells depleted of endogenous SNX5/6/32 that express wt or mutant versions of SNX5 to verify the conclusions from transient overexpression and qualitative imaging. Moreover, do the mutations in the IncE binding interface equally hamper recruitment of SNX6 and SNX32, similar to what is seen for SNX5? Are other endogenous SNX proteins co-recruited to the inclusion in this case?

2) Figure 3 would benefit from rearrangement and clearer labels. The authors may consider splitting panel C into three separate panels that are the easier to refer to in the main text. Moreover, critical amino acids delineating the SNX5 interface for IncE binding should be highlighted: Yellow labels of amino acids are difficult to see. Moreover, a different view of the interface from another angle may help to bring the message across. In 3C, maybe remove the non-relevant background and shades. The orientation of the magnifications relative to Figure 3 is not clear; for better orientation, indicate the deduced membrane binding site of the PX domain.

3) Regarding the role of PI3P in SNX5 recruitment: The text in the Results section refers to Figure 1—figure supplement 2, which is missing from the PDF. As for the data shown in Figure 4 please re-investigate the role of endosomal PI3P by quantitative image analysis and application of available selective Vps34 inhibitors (e.g. Vps34-IN1). Finally, split and merged channels as well as blow-ups should be shown in the actual figure to bring across this important point.

4) In Table 1 and Table 2*K_d_*s are given as μm concentrations, whereas in Figure 2 molar concentrations are plotted. This is somewhat confusing. I suggest to use μm throughout.

Figure 5: It is difficult to recognize the modeled head group. Better use a space filling representation; Figure 5 should be made clearer to help the reader compare the presence and absence of a phosphoinositide binding site.

Please include magnification bars in all micrographs; the word data is plural.

Results section, third paragraph, should refer to Figure 1, not 1G.

Figure 2 – Instead of affinity constants, better indicate *K_d_*s, which are more familiar to the casual reader.

Results section, Figure 3 – Fusion protein

The fusion protein should be more precisely described in the Results and Materials and Methods. How long is the linker, is the connection between C-terminus of PX domain and N-terminus of IncE long enough to account for the connection? Is the linker still intact upon crystallization? Are residues of the linker resolved in the crystal structure? Include the linker as dotted line in Figure 3.

Figure 4 in referred to as Figure 4 in the figure legend, and Figure 4 as Figure 4.

Does the superimposition in Figure 3—figure supplement 1B really include all 8 independent complexes found in the three crystal forms? It looks like only three complexes were superimposed.

Table 3: Average B-factor is missing for PDB 5TG1.

Table 3: Why is there such a huge gap in *R_work_/R_free_* for 5TGJ. Is this model fully refined?

[Editors' note: further revisions were requested prior to acceptance, as described below.]

Thank you for resubmitting your work entitled "Structural basis for the hijacking of endosomal sorting nexin proteins by *Chlamydia trachomatis*" for further consideration at *eLife*. Your revised article has been favorably evaluated by Vivek Malhotra (Senior editor) and three reviewers, one of whom is a member of our Board of Reviewing Editors. Although *eLife* tries very hard not to ask for extra, unnecessary experiments, the reviewers did not feel that you had adequately address their concerns about the presentation and quantification of the images as described below. We would like to be able to publish this story when you are able to address these issues satisfactorily and will welcome one last round of revision before we reach a binding decision.

Reviewer #1

Unfortunately, the authors have really not adequately improved the quality or the quantification of the micrographs. I strongly suggest that the micrographs be shown as individual figures with larger images, and we need to know how many cells were counted and how the analyses were carried out before we can publish the story. I hate to ask for more work but the images are an important part of the overall proposal.

Not sure how the authors came up with number of structures with 3 significant figures. The quantitative cell biological analyses have not been described.

Typo in in Introduction section:

“A fifth protein SNX32 is highly similar to SNX5 and 'SNX6'.”

Reviewer #3

I have to say that I remain unsatisfied with the way the authors have addressed the questions regarding the binding properties and cellular localization of mutant SNX5. All they have done is adding some Mander's colocalization coefficient, but it is unclear from how many cells and how many independent experiments these are derived. More importantly, I would still like to see formal proof that the F136A mutation does not impair SNX5-SNX1 (or 2) heterodimer formation in vitro and its localization at endosomes in mammalian cells assayed quantitatively. This in my view is required to support the corresponding statement in the text. Also, some better high mag colocalization images illustrating SNX5 WT and mutant targeting to endosomes would help to convince readers of the paper.

Likewise, my comment (#5 in the original review) regarding quantitative image analysis regarding effects of PI3K inhibition has not been addressed really.

[Editors' note: further revisions were requested prior to acceptance, as described below.]

Thank you for resubmitting your work entitled "Structural basis for the hijacking of endosomal sorting nexin proteins by *Chlamydia trachomatis*" for further consideration at *eLife*. Your revised article has been favorably evaluated by Vivek Malhotra (Senior editor), a Reviewing editor, and an additional reviewer. You will be pleased to learn that the reviewers found the revised manuscript to be much improved, and if you are able to make the following minor corrections, they have recommended in favor of publication in *eLife*.

The remaining issues that need to be addressed before acceptance, are as follows:

It appears that the statistical analysis of protein co-localization in Figure 1—figure supplement 2 and Figure 6—figure supplement 1 is based on the number of cells analyzed: 10 cells per condition – a fairly low number – which the authors say to derive from 2 independent experiments. According to convention in experimental biology individual cells from a cell line cannot be considered n for statistical analysis. Hence, n in fact equals "2" in this case, which does not allow for statistical testing by either parametric or non-parametric tests. However, as the phenotypes are clear-cut and in the interest of timely publication we recommend simply eliminating the statistical test from the figure, legend, and the text and simply leave the bar diagrams including error bars as a means of quantitation.

A last minor point: The heading of Figure 5 should read "Mutations […] formation in vitro" (e.g. delete "and in cells") as no cell data are contained in the respective figure. Provided these small changes we recommend acceptance and congratulate the authors on this nice piece of work.

---

## [Author Response]

*This manuscript reports the crystal structure of IncE C-terminal residues in complex with the PX domain of SNX5; in addition, it presents a protein binding interface for a PX domain in a subfamily of PX domain proteins that does not contain a functional phosphoinositide binding domain. Comparable binding affinity is seen for SNX 5, 6 and 32 PX domains but not that of SNX1. The story is compact and of high quality; it is an interesting variation on a theme of Chlamydial hijacking and the reviewers support presentation in eLife because it highlights an interesting biological interaction of medical importance: the elucidation of the molecular basis for hijacking of endosomal SNX5/6 and possibly 32 by bacterial IncE is important and of interest to the readers of eLife. Overall, the data support the main conclusions of the paper, however a few aspects should be strengthened by a few straightforward, additional experiments.*

*1) The authors provide qualitative data (e.g. Figure 4) from overexpression experiments in Chlamydia infected HeLa cells as an assay for IncE-mediated GFP-SNX5 recruitment. The staining in Figure 4 is hard to see and quantitative image analysis from multiple experiments will greatly strengthen the present story.*

We have provided Mander’s coefficients to measure the relative distribution of the GFP-SNX5 and mutant to the inclusion membrane as defined by mCherry-Rab25. This data has been added to the manuscript. With respect to “quantification” of the changes that occurs in the localization of IncE in Figure 4 and Figure 4—figure supplement 1 we do not consider any of the correlative methods appropriate for the following reasons. Firstly, the correlation of the IF signal for EEA1 and IncE is relatively low. They tend to be proximal to each other presumably as they are located on distinct subdomains of same endosomes (EEA1 – body; IncE/SNX5 – tubular domains). Secondly, the dramatic change of the mutant proteins to the cytoplasm results in a low level of correlation between endosome-associated proteins and the surrounding cytoplasm. We did calculate the number of punctate structures/per cell positive for GFP-IncE in an attempt to quantify any change in endosome association. For example, 29.556 +/- 4.558 punctate structures per cell were observed for IncE 91-132 while 0 +/- 0 for IncE 91-132 (F116D). However, as the analysis of the mutant proteins showed no punctate structures we felt documentation of this obvious difference provided little additional insight.

*The presence of endogenous endosomal SNX proteins that may interfere with recruitment and result in additional effects (e.g. as they may heterodimerize) that make the results difficult to interpret. Thus it would be worth testing IncE-mediated recruitment of SNX5 to bacterial inclusions using quantitative image analysis in HeLa cells depleted of endogenous SNX5/6/32 that express wt or mutant versions of SNX5 to verify the conclusions from transient overexpression and qualitative imaging.*

We requested some further clarification on this referee query from the editor, who stated “The reason the reviewer suggested this is that the mutation may affect dimerization and that's why the mutant SNX5 is not recruited to IncE inclusions. If the authors have other ways of addressing this point and are able to provide more quantitative data regarding the recruitment behavior of mutant SNX5 in cells this is fine.”

Our interpretation then is that the referees are concerned the mutant SNX5 (F136A) may also impact the folding and normal heterodimerisation of the SNX5 protein with either SNX1 or SNX2 Note that SNX5 does not homodimerise with itself, nor heterodimerise with SNX6 and SNX32 is not expressed endogenously in HeLa cells. We believe this concern is addressed by the fact that the SNX5 (F136A) mutant is recruited normally to endosomes (Figure 4). As this will only occur in a heterodimeric complex with either SNX1 or SNX2 we therefore conclude that the mutant is properly folded and otherwise functional.

Because we are not performing a rescue experiment to determine if mutation results in recovery of a ‘normal’ SNX5 function, we believe that performing the *C. trachomatis* inclusion recruitment experiment in a knockout background would not be informative. Here we are simply assessing if the binding deficient mutants are capable of being recruited to the chlamydial inclusion (SNX5) or endosome (GFP-IncE) through reciprocal interactions. As they are clearly not recruited (and see additional quantitation in Figure 4), it is not clear what additional information would be gained by also knocking out the endogenous proteins

*Moreover, do the mutations in the IncE binding interface equally hamper recruitment of SNX6 and SNX32, similar to what is seen for SNX5? Are other endogenous SNX proteins co-recruited to the inclusion in this case?*

To determine if mutation to the IncE binding interface also prevents interaction with SNX6 or SNX32 we performed additional in vitro experiments. Two new pieces of data clearly show that the IncE binding mechanism is identical in the three related SNX proteins. In new ITC data we find that mutation of the key Phe116 side chain in the IncE peptide blocks binding to both the SNX6 and SNX32 PX domains (new Figure 4—figure supplement 1), similarly to the interaction with SNX5 (Figure 4). In addition, in unpublished results we recently determined a crystal structure of the SNX32 PX domain in complex with the peptide from IncE (see image showing an overlay of SNX32 and SNX5 complexes). In this case the structure was solved with a co-complex including a synthetic peptide rather than a fusion protein as done for SNX5 (resolution, 2.3 Å; *R/R_free_* 0.19/0.23). We have not included this data in the manuscript, but it clearly confirms for the referees that the binding site is essentially identical in both proteins. In summary the IncE binding interface is highly conserved and we are certain that mutation of the strictly conserved Phe136 side- chain in all three SNX proteins would perturb IncE binding and inclusion recruitment.

Within cells we did not provide any experimental data that expression of the fragment of IncE “hampered recruitment” of SNX proteins to the bacterial inclusion. In fact, expression of these fragments does not have any impact on recruitment of endogenous SNX5 to the inclusion as the full-length membrane anchored bacterially synthesized IncE is still present on the outer membrane of the inclusion.

*2) Figure 3 would benefit from rearrangement and clearer labels. The authors may consider splitting panel C into three separate panels that are the easier to refer to in the main text. Moreover, critical amino acids delineating the SNX5 interface for IncE binding should be highlighted: Yellow labels of amino acids are difficult to see. Moreover, a different view of the interface from another angle may help to bring the message across. In 3C, maybe remove the non-relevant background and shades. The orientation of the magnifications relative to Figure 3 is not clear; for better orientation, indicate the deduced membrane binding site of the PX domain.*

This figure has now been adjusted to improve clarity as requested. The three panels have been separated into Figure 3 and referred to separately in the text and figure legend. The background shading in these close up images has been reduced and the yellow text has been changed to orange to make it easier to see. The two side chains critical for binding in SNX5 (Phe136) and IncE (Phe116) have been highlighted for the reader. We have decided not to indicate where the putative membrane-binding site is located in these images, as we feel this is better addressed in Figure 5 and would be confusing to include in this figure.

*3) Regarding the role of PI3P in SNX5 recruitment: The text in the Results section refers to Figure 1—figure supplement 2, which is missing from the PDF. As for the data shown in Figure 4 please re-investigate the role of endosomal PI3P by quantitative image analysis and application of available selective Vps34 inhibitors (e.g. Vps34-IN1). Finally, split and merged channels as well as blow-ups should be shown in the actual figure to bring across this important point.*

Figure 1—figure supplement 2 has been renamed as Video 1 to avoid confusion. It is not a PDF but an attached video showing the disruption of endosomal recruitment of GFP-SNX5 following wortmannin treatment in HeLa cells, while recruitment to the inclusion membrane is not affected. We have added an additional sentence to the legend of Figure 1—figure supplement 1 in order to clarify this. The figure panels have been split into separate channels with merged images shown alongside.

In the current study we have blocked formation of all 3-phosphoinositide species using the pan-specific PI3-kinase inhibitor wortmannin. As *C. trachomatis* inclusion recruitment of SNX- BAR proteins is not perturbed this indicates that the inclusion recruitment of SNX5 is likely driven by a protein-protein interaction (later shown to be IncE by SNX5 mutagenesis), rather than via 3- phosphoinositides including both the early and late-endosome enriched PtdIns3P and PtdIns(3,5)P2. We altered the text to emphasise wortmannin’s broad inhibitory effect to avoid confusion: “Interestingly, when infected cells are treated with wortmannin, a pan-specific inhibitor of phosphoinositide-3-kinase (PI3K) activity, we see a loss of the SNX proteins from punctate endosomes, but not from the inclusion membrane”. The reviewers suggest that we perform additional experiments using a specific inhibitor such as the Class III PI3K inhibitor Vps34-IN1, which will block specific formation of PtdIns3P. As wortmannin has no effect on inclusion recruitment, Vps34-IN1 is unlikely to provide additional insights.

*4) In Table 1 and Table 2 K_d_s are given as μm concentrations, whereas in Figure 2 molar concentrations are plotted. This is somewhat confusing. I suggest to use μm throughout.*

Figure 2 has been adjusted so that the axis is now plotted in μM as requested.

*Figure 5: It is difficult to recognize the modeled head group. Better use a space filling representation; Figure 5 should be made clearer to help the reader compare the presence and absence of a phosphoinositide binding site.*

Both Figure 5 and Figure 5 have now been modified as requested.

*Please include magnification bars in all micrographs; the word data is plural.*

Scale bars are now provided for all images.

*Results section, third paragraph, should refer to Figure 1, not 1G.*

This has been corrected.

*Figure 2 – Instead of affinity constants, better indicate K_d_s, which are more familiar to the casual reader.*

This has been altered in the figure so that the x-axis is now indicated as *1/K_d_* (M-1). It is clearer to the lay reader to show the axis in this orientation so that the ‘best’ binders are the highest bars in the graphs.

*Results section, Figure 3 – Fusion protein*

*The fusion protein should be more precisely described in the Results and Materials and methods. How long is the linker, is the connection between C-terminus of PX domain and N-terminus of IncE long enough to account for the connection? Is the linker still intact upon crystallization? Are residues of the linker resolved in the crystal structure? Include the linker as dotted line in Figure 3.*

We have included additional panels in Figure 3—figure supplement 1 that show the precise fusion protein sequence used for crystallization, as well as the highlighting the distances between the SNX5 C-terminus and the IncE N-terminus visible in our crystal structures. In each structure there are approximately 3-4 residues that are disordered and are not visible in the electron density, but the distances between the SNX5 C-termini and IncE N-termini are consistent with the expected linker lengths.

*Figure 4 in referred to as Figure 4 in the figure legend, and Figure 4 as Figure 4.*

The figure panels have been corrected.

*Does the superimposition in Figure 3—figure supplement 1B really include all 8 independent complexes found in the three crystal forms? It looks like only three complexes were superimposed.*

Yes it includes the 2 complexes from the P2_1_2_1_2_1_ crystal form, 2 complexes from the I2 crystal form and the 4 complexes from the P32 crystal form. The close alignment of the eight separate complexes makes is a consequence of the close structural similarity. The different conformations for some side-chains of the IncE peptides in Figure 3—figure supplement 1 provide the clearest display of the eight overlaid structures.

*Table 3: Average B-factor is missing for PDB 5TG1.*

This has been added to Table 3.

Table 3: Why is there such a huge gap in R_work_/R_free_ for 5TGJ. Is this model fully refined?

The model has been re-refined using the Form 1 crystal structure as a weak restraint in Phenix. This has resulted in an improvement in the *R/R_free_* from 0.185/0.267 to 0.199/0.242. The new structural statistics are included in Table 3.

[Editors' note: further revisions were requested prior to acceptance, as described below.]

*Reviewer #1*

*Unfortunately, the authors have really not adequately improved the quality or the quantification of the micrographs. I strongly suggest that the micrographs be shown as individual figures with larger images, and we need to know how many cells were counted and how the analyses were carried out before we can publish the story. I hate to ask for more work but the images are an important part of the overall proposal.*

As suggested by the reviewer we have now adjusted all immunofluorescence images so that they are single large figures for clarity. Immunofluorescence images in Figure 6 showing the effect of SNX5 and IncE mutagenesis have been recaptured, and we have provided details of the number of cells used in these analyses and more details related to the methods of quantification.

*Not sure how the authors came up with number of structures with 3 significant figures. The quantitative cell biological analysis have not been described.*

Mander’s correlation coefficients are now reported to 2 significant figures. As discussed above, details are now provided in the Materials and methods section and figure legends on the quantitative cell biology analyses.

*Typo in in Introduction section:*

“A fifth protein SNX32 is highly similar to SNX5 and 'SNX6'“

This typo has been corrected (same typo was found by reviewer 3).

*Reviewer #3*

*I have to say that I remain unsatisfied with the way the authors have addressed the questions regarding the binding properties and cellular localization of mutant SNX5. All they have done is adding some Mander's colocalization coefficient, but it is unclear from how many cells and how many independent experiments these are derived.*

Please see response to Reviewer 1 above.

*More importantly, I would still like to see formal proof that the F136A mutation does not impair SNX5-SNX1 (or 2) heterodimer formation* in vitro *and its localization at endosomes in mammalian cells assayed quantitatively. This in my view is required to support the corresponding statement in the text. Also, some better high mag colocalization images illustrating SNX5 WT and mutant targeting to endosomes would help to convince readers of the paper.*

As discussed above, GFP-SNX5 wild-type and mutant are both localized to endosomal puncta as normal. In additional data we have performed co-immunoprecipitation of these GFP-SNX5 constructs and find that the F136A mutation does not affect the heterodimeric interaction with endogenous SNX1 as we predicted (Figure 6—figure supplement 1). Overall we believe the data is conclusive that the F136A mutation has a specific affect on IncE interaction and inclusion membrane recruitment without perturbing other functions or folding of the SNX5 protein.

*Likewise, my comment (#5 in the original review) regarding quantitative image analysis regarding effects of PI3K inhibition has not been addressed really.*

We have now further addressed the original comment by Reviewer 2 by both using the specific Vps34 PI3K inhibitor Vps34-IN1, as well as quantification of the loss of endosomal localization following depletion of 3-phosphoinositides. Pearson’s correlation coefficients have been calculated for the endosomal colocalization of SNX1, SNX2 and SNX5 with endogenous endosomal markers. This data has been added to Figure 2—figure supplement 2. The text has been modified in the results as follows: “Interestingly, when infected cells are treated with wortmannin, a pan-specific inhibitor of phosphoinositide-3-kinase (PI3K) activity, we see a loss of the SNX proteins from punctate endosomes, but not from the inclusion membrane (Figure 1—figure supplement 1; Video 1). A similar result is seen for specific inhibition of PtdIns3P production by Vps34 using Vps34-IN1.”

[Editors' note: further revisions were requested prior to acceptance, as described below.]

*The remaining issues that need to be addressed before acceptance, are as follows:*

*It appears that the statistical analysis of protein co-localization in Figure 1—figure supplement 2 and Figure 6—figure supplement 1 is based on the number of cells analyzed:10 cells per condition – a fairly low number – which the authors say to derive from 2 independent experiments. According to convention in experimental biology individual cells from a cell line cannot be considered n for statistical analysis. Hence, n in fact equals "2" in this case, which does not allow for statistical testing by either parametric or non-parametric tests. However, as the phenotypes are clear-cut and in the interest of timely publication we recommend simply eliminating the statistical test from the figure, legend, and the text and simply leave the bar diagrams including error bars as a means of quantitation.*

We have amended the figures and figure legends, as well as the text in Materials and methods related to statistical testing.

*A last minor point: The heading of Figure 5 should read "Mutations […] formation* in vitro*" (e.g. delete "and in cells") as no cell data are contained in the respective figure. Provided these small changes we recommend acceptance and congratulate the authors on this nice piece of work.*

The figure legend has been corrected, and thanks for pointing out the mistake.